# DRAMA: MAMBA-ENABLED MODEL-BASED REINFORCEMENT LEARNING IS SAMPLE AND PARAMETER EFFICIENT

**Wenlong Wang, Ivana Dusparic, Yucheng Shi, Ke Zhang & Vinny Cahill**
School of Computer Science and Statistics
Trinity College Dublin, the University of Dublin
College Green, Dublin 2, Ireland
`{wangw1,ivana.dusparic,shiy2,zhangk2,vjcahill}@tcd.ie`

## ABSTRACT

Model-based reinforcement learning (RL) offers a solution to the data inefficiency that plagues most model-free RL algorithms. However, learning a robust world model often requires complex and deep architectures, which are computationally expensive and challenging to train. Within the world model, sequence models play a critical role in accurate predictions, and various architectures have been explored, each with its own challenges. Currently, recurrent neural network (RNN)-based world models struggle with vanishing gradients and capturing long-term dependencies. Transformers, on the other hand, suffer from the quadratic memory and computational complexity of self-attention mechanisms, scaling as $O(n^2)$, where $n$ is the sequence length.

To address these challenges, we propose a state space model (SSM)-based world model, **Drama**, specifically leveraging Mamba, that achieves $O(n)$ memory and computational complexity while effectively capturing long-term dependencies and enabling efficient training with longer sequences. We also introduce a novel sampling method to mitigate the suboptimality caused by an incorrect world model in the early training stages. Combining these techniques, Drama achieves a normalised score on the *Atari100k* benchmark that is competitive with other state-of-the-art (SOTA) model-based RL algorithms, using only a 7 million-parameter world model. Drama is accessible and trainable on off-the-shelf hardware, such as a standard laptop. Our code is available at `https://github.com/realwenlongwang/Drama.git`.

## 1 INTRODUCTION

Deep Reinforcement Learning (RL) has achieved remarkable success in various challenging applications, such as Go (Silver et al., 2016; 2017), Dota (Berner et al., 2019), Atari (Mnih et al., 2013; Schrittwieser et al., 2020), and MuJoCo (Schulman et al., 2017; Haarnoja et al., 2018). However, training policies capable of solving complex tasks often requires millions of environment interactions, which can be impractical and pose a barrier to real-world applications. Thus, improving sample efficiency has become a critical goal in RL algorithm development.

World models have shown promise in improving sample efficiency by generating artificial training samples through an autoregressive process, a method referred to as model-based RL (Micheli et al., 2023; Robine et al., 2023; Zhang et al., 2023; Hafner et al., 2023). In this approach, interaction data is used to learn the environment dynamics using a sequence model, allowing the agent to train on artificial experiences generated by the resulting sequence model instead of relying on real-world interactions. This approach shifts the problem from improving the policy directly using real samples (which is sample inefficient) to improving the accuracy of the world model to match the real environment (which is more sample efficient). However, model-based RL faces a well-known challenge: when the model is inaccurate due to limited observed samples, especially early in training, the policy eventually learned can converge to suboptimal behaviour, and detecting model errors is difficult, if not impossible.

In sequence modelling, linear complexity (in sequence length) is highly desirable because it allows models to efficiently process longer sequences without a dramatic increase in computational and memory resources. This is particularly important when training world models, which require efficient sequence modelling to simulate complex environments over long time horizons. RNNs, particularly advanced variants like Long Short-Term Memory (LSTM) and Gated Recurrent Units (GRU), offer linear complexity, making them computationally attractive for this task. However, RNNs still struggle with vanishing gradient issues and exhibit limitations in capturing long-term dependencies (Hafner et al., 2019; 2023). More recently, transformer architectures, which have dominated natural language processing (NLP) (Vaswani et al., 2017), have gained traction in fields like image processing and offline RL following groundbreaking work in these areas (Dosovitskiy et al., 2021; Chen et al., 2021). The transformer structure has demonstrated its effectiveness in model-based RL as well (Micheli et al., 2023; Robine et al., 2023; Zhang et al., 2023). However, transformers suffer from both memory and computation complexity that scale as $O(n^2)$, where $n$ is the sequence length, posing challenges for world models that require long sequences[1] to simulate complex environments.

Currently, SSMs are attracting significant attention for their ability to efficiently model long-sequence problems with linear complexity. Among SSMs, Mamba has emerged as a competitive alternative to transformer-based architectures in various fields, including NLP (Gu & Dao, 2024; Dao & Gu, 2024), computer vision (Zhu et al., 2024), and offline RL (Lv et al., 2024). Applying Mamba's architecture to model-based RL is particularly appealing due to its linear memory and computational scaling with sequence length, coupled with its ability to capture long-term dependencies effectively. Moreover, efficiently capturing environmental dynamics can reduce the likelihood that the behaviour policy being learned within an inaccurate world model, a challenge we further address by incorporating a novel dynamic frequency-based sampling method. In this paper, we make three key contributions:

- We introduce Drama, the first model-based RL agent built on the Mamba SSM, with Mamba-2 as the core of its architecture. We evaluate Drama on the Atari100k benchmark, demonstrating that it achieves performance comparable to other SOTA algorithms while using only a 7 million trainable parameter world model.

- We compare the performance of Mamba and Mamba-2 , demonstrating that Mamba-2 achieves superior results as a sequence model in the Atari100k benchmarks, despite slightly limiting expressive power to enhance training efficiency.

- Finally, we propose a novel but straightforward sampling method, dynamic frequency-based sampling (DFS), to mitigate the challenges posed by imperfect sequence models.

## 2 METHOD

We describe the problem as a Partially Observable Markov Decision Process (POMDP), where at each discrete time step $t$, the agent observes a high-dimensional image $\mathbf{O}_t \in \mathbb{O}$ rather than the true state $s_t \in \mathbb{S}$ with the conditional observation probability given by $p(\mathbf{O}_t|s_t)$. The agent selects actions from a discrete action set $a_t \in \mathbb{A} = \{0, 1, \ldots, n\}$. After executing an action $a_t$, the agent receives a scalar reward $r_t \in \mathbb{R}$, a termination flag $e_t \in [0, 1]$, and the next observation $\mathbf{O}_{t+1}$. The dynamics of the MDP are described by the transition probability $p(s_{t+1}, r_t|s_t, a_t)$. The behaviour of the agent is determined by a policy $\pi(\mathbf{O}_t; \boldsymbol{\theta})$, parameterised by $\boldsymbol{\theta}$, where $\pi : \mathbb{O} \rightarrow \mathbb{A}$ maps the observation space to the action space. The goal of this policy is to maximise the expected sum of discounted rewards $\mathbb{E} \sum_t \gamma^t r_t$, given that $\gamma$ is a predefined discount factor.

Unlike model-free RL, model-based RL does not rely directly on real experiences to improve the policy $\pi(\mathbf{O}_t; \boldsymbol{\theta})$ (Sutton & Barto, 1998). There are various approaches to obtaining a world model, including Monte Carlo tree search (Schrittwieser et al., 2020), offline imitation learning (DeMoss et al., 2023) and latent sequence models (Hafner et al., 2019). In this work, we focus on learning a world model $f(\mathbf{O}_t, a_t; \omega)$ from actual experiences to capture the dynamics of the POMDP in a latent space. The actual experiences are stored in a replay buffer, allowing them to be repeatedly sampled for training the world model. The world model consists of a variational autoencoder (VAE) (Kingma

---

[1]According to (Tay et al., 2021), a long sequence is defined as having a length of 1,000 or more.

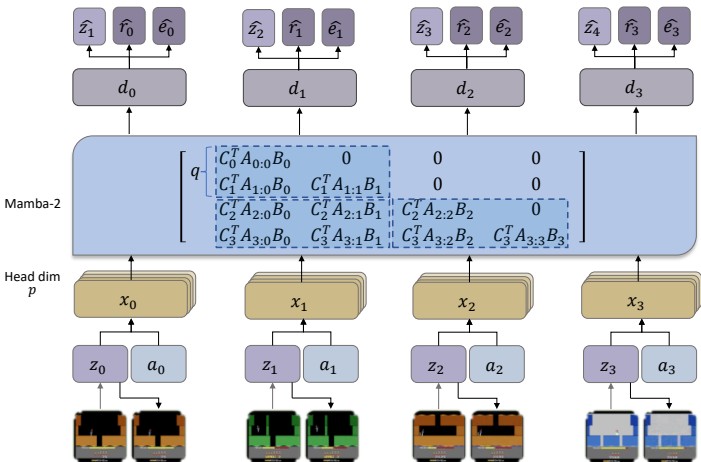

Figure 1: Drama world model architecture. At each sequence index $t$, the raw game frames are encoded into $z_t$ and combined with the action $a_t$ as input to the Mamba blocks. The input channel dimension is divided by the head dimension $p$ to generate the deterministic recurrent state $d_t$. This recurrent state $d_t$ is used to predict the next embedding $\hat{z}_{t+1}$, reward $\hat{r}_t$, and termination flag $\hat{e}_t$, which represent the outcomes based on the current frame and action. The decoder reconstructs the original frame from the encoded embeddings $z_t$ rather than from the predicted embeddings $\hat{z}_t$. The Mamba-2 block employs a semi-separable matrix structure, which can be decomposed into $q \times q$ sub-matrices, enabling more efficient computation and processing.

& Welling, 2014; Hafner et al., 2021), a sequence model, and linear heads to predict rewards and termination flags. The details of our world model are discussed in Section 2.2.

After each update to the world model, a batch of experiences is sampled from the replay buffer to initiate a process called 'imagination'. Starting from an actual initial observation and using an action generated by the current behaviour policy, the sequence model generates the next latent state. This process is repeated until the agent collects sufficient imagined samples for policy improvement. We explain this process in detail in Section 2.3.

## 2.1 STATE SPACE MODELLING WITH MAMBA

SSMs are mathematical frameworks inspired by control theory to represent the complete state of a system at a given point in time. These models map an input sequence to an output sequence $\boldsymbol{x} \in \mathbb{R}^l \to \boldsymbol{y} \in \mathbb{R}^l$, where $l$ denotes the sequence length. In structured SSMs, a hidden state $\boldsymbol{H} \in \mathbb{R}^{(n,l)}$ is used to track the sequence dynamics, as described by the following equations:

$$
\begin{aligned}
\boldsymbol{H}_t &= \boldsymbol{A}\boldsymbol{H}_{t-1} + \boldsymbol{B}x_t \\
y_t &= \boldsymbol{C}^\mathsf{T}\boldsymbol{H}_t
\end{aligned}
\tag{1}
$$

where $\boldsymbol{A} \in \mathbb{R}^{(n,n)}, \boldsymbol{B} \in \mathbb{R}^{(n,1)}, \boldsymbol{C} \in \mathbb{R}^{(n,1)}$ and $\boldsymbol{H}_t \in \mathbb{R}^{(n,1)}$, in which $n$ represents the predefined dimension of the hidden state that remains invariant to the sequence length. To efficiently compute the hidden states, it is common to structure $\boldsymbol{A}$ as a diagonal matrix, as discussed in (Gu et al., 2022b; Gupta et al., 2022; Smith et al., 2023; Gu & Dao, 2024). Additionally, selective SSMs, such as Mamba, extend the matrices $(\boldsymbol{A}, \boldsymbol{B}, \boldsymbol{C})$ to be time-varying, introducing an extra dimension corresponding to the sequence length. The shapes of these time-varying matrices are $\mathbf{A} \in \mathbb{R}^{(T,N,N)}, \boldsymbol{B} \in \mathbb{R}^{(T,N)}$, and $\boldsymbol{C} \in \mathbb{R}^{(T,N)}$ [2].

---

[2] In Mamba, the time variation of $\boldsymbol{A}$ is influenced by a discretisation parameter $\Delta$. For more details, please refer to (Gu & Dao, 2024)

Dao & Gu (2024) introduced the concept of structured state space duality (SSD), which further restricts the diagonal matrix $\boldsymbol{A}$ to be a scalar multiple of the identity matrix, forcing all diagonal elements to be identical. To address the resulting reduced expressive power, Mamba-2 introduces a multi-head technique, akin to attention, by treating each input channel as $p$ independent sequences. Unlike Mamba, which computes SSMs as a recurrence, Mamba-2 approaches the sequence transformation problem through matrix multiplication, which is more GPU-efficient:

$$
\begin{aligned}
y_t &= \boldsymbol{C}_t^{\mathsf{T}} \boldsymbol{H}_t \\
y_t &= \sum_{i=0}^{t} \boldsymbol{C}_t^{\mathsf{T}} \boldsymbol{A}_{t:i} \boldsymbol{B}_i x_i
\end{aligned}
\tag{2}
$$

where $\boldsymbol{A}_{t:i}$ is $\boldsymbol{A}_t \boldsymbol{A}_{t-1} \ldots \boldsymbol{A}_{i+1}$. This allows the SSM to be formulated as a matrix transformation:

$$
\boldsymbol{y} = SSM(\boldsymbol{x}; \mathsf{A}, \boldsymbol{B}, \boldsymbol{C}) = \boldsymbol{M}x
$$
$$
M_{j,i} \coloneqq \begin{cases} \boldsymbol{C}_t^{\mathsf{T}} \boldsymbol{A}_{t:i} \boldsymbol{B}_i & \text{if } j \geq i \\ 0 & \text{if } j < i \end{cases}
\tag{3}
$$

Mamba-2 reformulates the state-space equations as a single matrix multiplication using semi-separable matrices (Vandebril et al., 2005; Dao & Gu, 2024), which is well known in computational linear algebra, as shown by Figure 1. The matrix $\boldsymbol{M}$ can also be written as:

$$
\boldsymbol{M} = \boldsymbol{L} \circ \boldsymbol{C} \boldsymbol{B}^{\mathsf{T}} \in \mathbb{R}^{(T,T)}
$$
$$
\boldsymbol{L} = \begin{bmatrix} 1 & & & & \\ a_1 & 1 & & & \\ a_2 a_1 & a_2 & 1 & & \\ \vdots & \vdots & \ddots & \ddots & \\ a_{\mathsf{T}-1} \ldots a_1 & a_{\mathsf{T}-1} \ldots a_2 & \ldots & a_{\mathsf{T}-1} & 1 \end{bmatrix}
\tag{4}
$$

where $a_t \in [0, 1]$ is an input-dependent scalar. The matrix $\boldsymbol{L}$ bridges the SSM mechanism with the causal self-attention mechanism by removing the softmax function and applying a mask matrix $\boldsymbol{L}$ to the 'attention-like' matrix. It is equivalent to causal linear attention when all $a_t = 1$. As a result, Mamba-2 achieves 2-8 times faster training speeds than Mamba, while maintaining linear scaling with sequence length.

## 2.2 WORLD MODEL LEARNING

Our world model has two main components: an autoencoder and a sequence model. Additionally it includes two MLP heads for reward and termination predictions. The architecture of the world model is illustrated in Figure 1.

### 2.2.1 DISCRETE VARIATIONAL AUTOENCODER

The autoencoder extends the standard variational autoencoder (VAE) architecture (Kingma & Welling, 2014) by incorporating a fully-connected layer to discretise the latent embeddings, consistent with previous approaches (Hafner et al., 2021; Robine et al., 2023; Zhang et al., 2023). The raw observation is a three-dimensional image, $\mathbf{O}_t \in [0, 255]^{(h,w,c)}$, at time step $t$. The encoder compresses the observation into a discrete latent vector, denoted as $\boldsymbol{z}_t \sim p(\boldsymbol{z}_t | \mathbf{O}_t)$. The decoder reconstructs the raw image, $\hat{\mathbf{O}}_t$, from $\boldsymbol{z}_t$. Gradients are passed directly from the decoder to the encoder using the straight-through estimator, bypassing the sampling operation during backpropagation (Bengio et al., 2013).

### 2.2.2 SEQUENCE MODEL

The sequence model simulates the environment in the latent variable space, $\boldsymbol{z}_t$, using a deterministic state variable, $\boldsymbol{d}_t$. Note that this is distinct from the hidden states typically used by SSMs, like Mamba and Mamba-2, to track dynamics. At each time step $t$, the next token in the sequence is determined by both the current latent variable, $\boldsymbol{z}_t$ and the current action $a_t$. To integrate these,

we concatenate them and project the result using a fully connected layer before passing it to the sequence model. Given a sequence length $l$, the deterministic state is derived from all previous latent variables and actions. The sequence model can be expressed as:

$$
\begin{aligned}
\text{Seuqnce model:} & \quad \boldsymbol{d}_t = f(\boldsymbol{z}_{t-l:t}, a_{t-l:t}; \omega) \\
\text{Latent variable predictor:} & \quad \hat{\boldsymbol{z}}_{t+1} \sim p(\hat{\boldsymbol{z}}_{t+1} | \boldsymbol{d}_t; \omega)
\end{aligned}
\tag{5}
$$

We implement the sequence model with Mamba-2 (Dao & Gu, 2024). Specifically, each time a batch of samples, denoted as $\mathbf{O} \in [0, 255]^{(b,l,h,w,c)}$, is drawn from the experience buffer $\mathcal{E}$, where $b$ is the batch size, $l$ the sequence length, and $h, w, c$ the image height, width, and channel dimension respectively. After encoding, the batch will be compressed to $\mathbf{Z} \in \mathbb{R}^{(b,l,d)}$ where $d$ is the dimension of the latent variable. The latent variable passes through a linear layer with the action to produce the input $\mathbf{X} \in \mathbb{R}^{(b,l,d)}$ of the Mamba blocks. To fully leverage GPU parallelism, the training process must strictly avoid sequential dependencies. That is, at time step $t$, the sequence model predicts the latent variable $\hat{\boldsymbol{z}}_{t+1}$, and its target $\boldsymbol{z}_{t+1}$ depends solely on the observation $\mathbf{O}_{t+1}$ as shown in Figure 1. Unlike DreamerV3, where $\boldsymbol{z}_{t+1} \sim p(\boldsymbol{z}_{t+1}|\mathbf{O}_{t+1}, \boldsymbol{d}_t)$, this approach eliminates sequential dependence.

Mamba processes the input tensor $\mathbf{X}_{b,:l,d}$ into a sequence of hidden states $\mathbf{H} \in \mathbb{R}^{(b,l-1,n)}$, which are then mapped back to the deterministic state sequence $\mathbf{D}_{b,:l,d}$ using time-varying parameters. Since the hidden states operate in a fixed dimension $n$ (unlike standard attention mechanisms, where the state scales with the sequence length), Mamba achieves linear computational complexity in $l$.

Mamba-2 applies a similar transformation but leverages matrix multiplication. The input tensor $\mathbf{X}$'s dimension $d$ is first split into $d/p$ heads, which are processed independently. The transformation matrix is a specially designed semiseparable lower triangular matrix, which can be decomposed into $q \times q$ blocks. Specialised blocks handle causal attention over short ranges and hidden state transformations, enabling efficient GPU computation.

## 2.3 BEHAVIOUR POLICY LEARNING

The behaviour policy is trained within the 'imagination', an autoregressive process driven by the sequence model. Specifically, a batch of $b_{img}$ trajectories, each of length $l_{img}$, is sampled from the replay buffer. Leveraging Mamba's efficiency with long sequences , we use real-world transitions to estimate a more informative hidden state for the 'imagination' process. Rollouts begin from the last transition of each sequence (at step $l_{img}$) and continue for $h$ steps. Notably, the rollout does not stop when an episode ends, unlike the prior SSM-based meta-RL model (Lu et al., 2023) where the hidden state must be manually reset, as the Mamba-based sequence model automatically resets the state at episode boundaries (Gu & Dao, 2024).

A key difference between Mamba- and transformer-based world models lies in the 'imagination' process: Mamba updates inference parameters independently of sequence length, accelerating the 'imagination' process, which is a major time-consuming phase in model-based RL. The behaviour policy's state concatenates the prior discrete variable $\hat{\boldsymbol{z}}_t$ with the deterministic variable $\boldsymbol{d}_t$ to exploit the temporal information. While the behaviour policy utilises a standard actor-critic architecture, other on-policy algorithms can also be applied. In this work, we adopt the recommendations from (Andrychowicz et al., 2021) and adjust the loss functions and value normalisation techniques as described in (Hafner et al., 2023).

## 2.4 DYNAMIC FREQUENCY-BASED SAMPLING (DFS)

In model-based RL, the behaviour model often underestimates rewards due to inaccuracies in the world model, impeding exploration and error correction (Sutton & Barto, 1998). These inaccuracies are particularly common early in training when the model is fitted to limited data. Thus, we propose a sample-efficient method to address this issue, i.e., Dynamic Frequency-based Sampling (DFS).

The primary objective is to sample transitions that the world model has sufficiently learned to ensure reliable 'imagination'. To accomplish this, we maintain two vectors during training, each matching the length of the transition buffer $|\mathcal{E}|$. For the world model, $\boldsymbol{v} = (v_1, v_2, \ldots, v_{|\mathcal{E}|})$, where $v_i \in \mathbb{Z}^+$ for $i \in \{1, 2, \ldots, |\mathcal{E}|\}$, which tracks the number of times a transition has been sampled to improve the world model. The resulting sampling probabilities are

computed as, $(p_1, p_2, \ldots, p_{|\mathcal{E}|}) = \texttt{softmax}(-\boldsymbol{v})$, similar to (Robine et al., 2023). For 'imagination', $\boldsymbol{b} = (b_1, b_2, \ldots, b_{|\mathcal{E}|})$, where $b_i \in \mathbb{Z}^+$ for $i \in \{1, 2, \ldots, |\mathcal{E}|\}$, which counts the times that the transition has been sampled to improve the behaviour policy. The sampling probabilities are denoted as, $(p_1, p_2, \ldots, p_{|\mathcal{E}|}) = \texttt{softmax}(f(\boldsymbol{v}, \boldsymbol{b}))$, where $f(\boldsymbol{v}, \boldsymbol{b}) = \boldsymbol{v} - \boldsymbol{b} - \max(0, \boldsymbol{v} - \boldsymbol{b})$. During training, two cases arise: 1) $\exists i \in |\mathcal{E}|$, $v_i \geq b_i$, $f(v_i, b_i) = 0$, In this case, the transition has been trained more frequently with the world model than with the behaviour policy, suggesting that the world model is likely capable of making accurate predictions from this transition. 2) $\exists i \in |\mathcal{E}|, v_i < b_i, f(v_i, b_i) = v_i - b_i$, signaling that the transition is either likely under-trained for the world model rollouts or overfitted to the behaviour policy. Consequently, the probability of selecting this transition for behaviour policy training decreases. These two mechanisms ensure that 'imagination' sampling favours transitions learned by the world model, while avoiding excessive determinism.

## 3 EXPERIMENTS

In this work, the proposed Drama framework is implemented on top of the STORM infrastructure (Zhang et al., 2023). We evaluate the model using the **Atari100k benchmark** (Kaiser et al., 2020), which is widely used for assessing the sample efficiency of RL algorithms. Atari100k limits interactions with the environment to 100,000 steps (equivalent to 400,000 frames with 4-frame skipping). We present the benchmark and analyse our results in Section 3.1 . Ablation experiments and their analysis are provided in Section 3.2.

### 3.1 ATARI100K RESULTS

We compare Drama against SOTA MBRL algorithms across 26 Atari games in the Atari100k benchmark. In Table 1, the 'Normalised Mean' refers to the average normalised score, calculated as: $(evaluated\_score - random\_score)/(human\_score - random\_score)$. For each game, we train Drama with 5 independent seeds and track training performance using a 5-episode running average, as recommended by Machado et al. (2018), a practice also followed in related work (Hafner et al., 2023).

Despite utilising an extra-small world model (7M parameters, referred to as the XS model), Drama achieves performance comparable to IRIS and TWM. To enable a like-for-like comparison between Drama and DreamerV3 with a similar number of parameters, we evaluate the learning curves of Drama and a 12M-parameter variant of DreamerV3 (referred to as DreamerV3XS) on the full Atari100K benchmark. As shown in Figure 4 in the appendix, Drama significantly outperforms DreamerV3XS, achieving a normalised mean score of 105 compared to 37 and a normalised median score of 27 compared to 7, as presented in Table 3.

Table 1 demonstrates that Drama, with Mamba-2 as the sequence model, is both sample- and parameter-efficient. For comparison, SimPLe (Kaiser et al., 2020) trains a video prediction model to optimise a PPO agent (Schulman et al., 2017), while SPR (Schwarzer et al., 2021) uses a sequence model to predict in latent space, enhancing consistency through data augmentation. TWM (Robine et al., 2023) employs a Transformer-XL architecture to capture dependencies among states, actions, and rewards, training a policy-based agent. This method incorporates short-term temporal information into the embeddings to avoid using the sequence model during actual interactions. Similarly, IRIS (Micheli et al., 2023) uses a Transformer as its sequence model, but generates new samples in image space, allowing pixel-level feature extraction for behaviour policies. DreamerV3 (Hafner et al., 2023), which employs an RNN-based sequence model along with robustness techniques, achieves superhuman performance on the Atari100k benchmark using a 200M parameter model—20 times larger than our XS model. STORM (Zhang et al., 2023), which adopts many of DreamerV3's robustness techniques while replacing the sequence model with a transformer, reaches similar performance on the Atari100k benchmark as DreamerV3.

Drama excels in games like `Boxing` and `Pong`, where the player competes against an autonomous opponent in simple, static environments, requiring a less intense autoencoder. This strong performance indicates that Mamba-2 effectively captures both ball dynamics and the opponent's position. Similarly, Drama performs well in `Asterix`, which benefits from its ability to predict object movements. However, Drama struggles in `Breakout`, where performance can be improved

| | Random | Human | PPO | SimPLe | SPR | TWM | IRIS | STORM | DreamerV3 | DramaXS |
|---|---|---|---|---|---|---|---|---|---|---|
| Alien | 228 | 7128 | 276 | 617 | 842 | 675 | 420 | 984 | **1118** | 820 |
| Amidar | 6 | 1720 | 26 | 74 | 180 | 122 | 143 | **205** | 97 | 131 |
| Assault | 222 | 742 | 327 | 527 | 566 | 683 | **1524** | 801 | 683 | 539 |
| Asterix | 210 | 8503 | 292 | 1128 | 962 | 1117 | 854 | 1028 | 1062 | **1632** |
| BankHeist | 14 | 753 | 14 | 34 | 345 | 467 | 53 | **641** | 398 | 137 |
| BattleZone | 2360 | 37188 | 2233 | 4031 | 14834 | 5068 | 13074 | 13540 | **20300** | 10860 |
| Boxing | 0 | 12 | 3 | 8 | 36 | 78 | 70 | 80 | **82** | 78 |
| Breakout | 2 | 30 | 3 | 16 | 20 | 20 | **84** | 16 | 10 | 7 |
| ChopperCommand | 811 | 7388 | 1005 | 979 | 946 | 1697 | 1565 | 1888 | **2222** | 1642 |
| CrazyClimber | 10780 | 35829 | 14675 | 62584 | 36700 | 71820 | 59324 | 66776 | **86225** | 83931 |
| DemonAttack | 152 | 1971 | 160 | 208 | 518 | 350 | **2034** | 165 | 577 | 201 |
| Freeway | 0 | 30 | 2 | 17 | 19 | 24 | 31 | **34** | 0 | 15 |
| Frostbite | 65 | 4335 | 127 | 237 | 1171 | 1476 | 259 | 1316 | **3377** | 785 |
| Gopher | 258 | 2412 | 368 | 597 | 661 | 1675 | 2236 | **8240** | 2160 | 2757 |
| Hero | 1027 | 30826 | 2596 | 2657 | 5859 | 7254 | 7037 | 11044 | **13354** | 7946 |
| Jamesbond | 29 | 303 | 41 | 100 | 366 | 362 | 463 | 509 | **540** | 372 |
| Kangaroo | 52 | 3035 | 55 | 51 | 3617 | 1240 | 838 | **4208** | 2643 | 1384 |
| Krull | 1598 | 2666 | 3222 | 2205 | 3682 | 6349 | 6616 | 8413 | 8171 | **9693** |
| KungFuMaster | 258 | 22736 | 2090 | 14862 | 14783 | 24555 | 21760 | **26183** | 25900 | 23920 |
| MsPacman | 307 | 6952 | 366 | 1480 | 1318 | 1588 | 999 | **2673** | 1521 | 2270 |
| Pong | -21 | 15 | -20 | 13 | -5 | **19** | 15 | 11 | -4 | 15 |
| PrivateEye | 25 | 69571 | 100 | 35 | 86 | 87 | 100 | **7781** | 3238 | 90 |
| Qbert | 164 | 13455 | 317 | 1289 | 866 | 3331 | 746 | **4522** | 2921 | 796 |
| RoadRunner | 12 | 7845 | 602 | 5641 | 12213 | 9109 | 9615 | 17564 | **19230** | 14020 |
| Seaquest | 68 | 42055 | 305 | 683 | 558 | 774 | 661 | 525 | **962** | 497 |
| UpNDown | 533 | 11693 | 1502 | 3350 | 10859 | 15982 | 3546 | 7985 | **46910** | 7387 |
| Normalised Mean (%) | 0 | 100 | 11 | 33 | 62 | 96 | 105 | 127 | 125 | 105 |
| Normalised Median (%) | 0 | 100 | 3 | 13 | 40 | 51 | 29 | 58 | 49 | 27 |

Table 1: Comparison of game performance metrics for various algorithms across multiple Atari games. For `Freeway` IRIS enhances exploration using a distinct set of hyperparameters, while STORM leverages offline expert knowledge. TWM reports the results with a 21.6M model while IRIS does not report the exact number of parameters, they use the same transformer embedding dimension and layer number as TWM plus a behaviour policy with CNN layers. DreamerV3 notably uses a 200M parameter model and achieves good results in a series of diverse tasks. STORM does not report the number of trainable parameters.

with a more robust autoencoder in Figure 6. Additionally, Drama excels in games like `Krull` and `MsPacman`, which require longer sequence memory, but faces challenges in sparse reward games like `Jamesbond` and `PrivateEye`.

## 3.2 ABLATION EXPERIMENTS

In this section, we present three ablation experiments to evaluate key components of Drama. First, we compare dynamic frequency-based sampling performance against uniform sampling on the full Atari100k benchmark, demonstrating its effectiveness across diverse environments. Secondly, we compare Mamba and Mamba-2 on a subset of Atari games, including `Krull`, `Boxing`, `Freeway`, and `Kangaroo`, to highlight the differences in their performance when applied to dynamic gameplay scenarios. Lastly, we compare the long-sequence processing capabilities of Mamba, Mamba-2, and GRU in a custom Grid World environment. This experiment focuses on a prediction task using different sequence models, offering insights into their sequence modelling capabilities, which are crucial for MBRL applications especially if long-sequence modelling is important.

### 3.2.1 DYNAMIC FREQUENCY-BASED SAMPLING

In this experiment, we compare DFS with the uniform sampling method in the Mamba-2-based Drama on the full Atari100k benchmark. In Table 4, DFS is more effective than the uniform sampling overall , achieving a 105% normalised mean score (vs. the uniform's 80%), despite both methods sharing similarly median performance (27% vs. 28%). As shown in Figure 5, DFS shows signif-

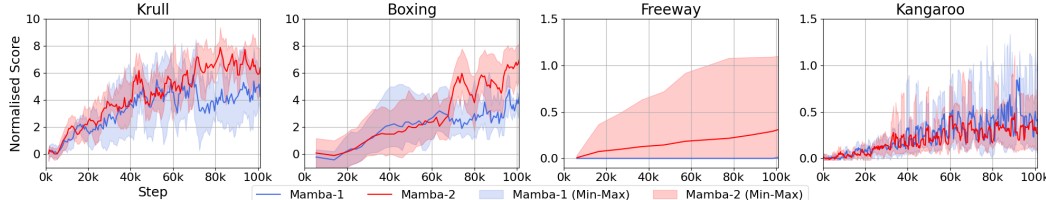

Figure 2: Mamba vs. Mamba-2. Mamba2 has shown a superior performance to Mamba in three out of four games. Both Mamba and Mamba-2 use DFS in this experiment.

icant advantages in games requiring adaptation to evolving dynamics, such as `Alien`, `Asterix`, `BankHeist`, and `Seaquest`. Additionally, DFS performs well in opponent-based games such as `Boxing` and `Pong`, where exploiting the weaknesses of the opponent AI is essential. However, DFS performs less effectively in games like `Breakout` and `KungFuMaster`, likely because the critical game dynamics are accessible early in the gameplay.

### 3.2.2 MAMBA VS. MAMBA-2

As mentioned in Sec 2.1, Mamba-2 imposes restrictions on the diagonal matrix $\boldsymbol{A}$ to improve efficiency. However, whether these restrictions degrade performance of SSMs remains unclear, as prior work lacks conclusive theoretical or empirical evidence (Dao & Gu, 2024). In response to this gap, we compare Mamba-2 and Mamba as the backbone of the world model in model-based RL. We conducted ablation experiments using DFS, with both architectures configured under identical hyperparameters.

Figure 2 illustrates that Mamba-2 outperforms Mamba in the games `Krull`, `Boxing` and `Freeway`. In `Krull`, the player navigates through different scenes and solves various tasks. In the later stages, rescuing the princess while avoiding hits results in a significant score boost, while failure leads to a plateau in score. As shown, Mamba experiences a score plateau in `Krull`, whereas Mamba-2 successfully overcomes this challenge, leading to higher performance. Note that `Freeway` is a sparse reward game requiring high-quality exploration. A positive training effect is achieved only by combining DFS with Mamba-2 without any additional configuration.

### 3.2.3 SEQUENCE MODELS FOR LONG-SEQUENCE PREDICTABILITY TASKS

To assess the efficiency of Mamba and Mamba-2 in long-range modelling compared to Transformers and GRUs, which are widely used in recent MBRL approaches, we present a simple yet representative grid world environment[3], as illustrated in Figure 3a. The learning objectives here are twofold: 1) the sequence model must reconstruct (predict) the correct grid-world geometry over a long sequence and 2) the sequence model must accurately generate the agent's location within the grid world, reflecting the prior sequence of movements. To achieve this, we represent a trajectory as a long sequence by flattening consecutive frames (row-wise tokenisation of frames) and separating each frame with a movement action $a$. Let the size of the grid world be $l_g$. Then, each frame can be tokenised into a sequence of length $l_f = l_g^2 + 1$, as depicted in Figure 3b. Since $l \gg l_f$, the task demands strong long-range sequence modelling to ensure geometric and logical consistency in predictions–a core requirement for MBRL sequence models.

We compare GRU, Transformer, Mamba, and Mamba-2 in this grid world environment, where $l_g = 5$ and $l_f = 26$, considering two sequence lengths: a short sequence length $l = 8 \times l_f$ and a long sequence length $l = 64 \times l_f$. Performance is measured via training time, memory usage and reconstruction error (lower time consumption and reconstruction error indicate better environment understanding). Results show that, Mamba and Mamba-2 achieve equivalent low error and short training time in both sequence lengths compared to other methods. However, Mamba-2 demonstrates the lowest training time over all methods. These findings confirm that the proposed Mamba-based architecture presents a strong capability to capture essential information, particularly in scenarios involving long sequence lengths.

---

[3]Implementation based on (Torres–Leguet, 2024)

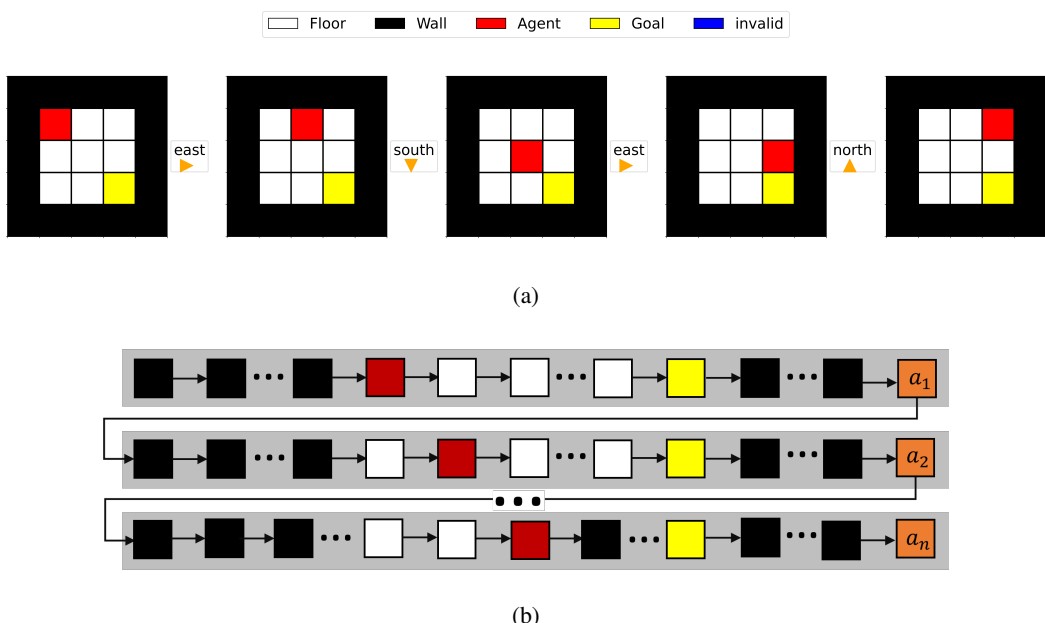

Figure 3: Illustrations of the grid world environment and its reconstruction into a sequential format. (a) Sequence of consecutive frames in the grid world environment. The Example presents a sequence of consecutive frames, arranged from left to right. Each frame represents a $5 \times 5$ grid, where the outer 16 cells are black walls, and the central $3 \times 3$ grid is the reachable space. The red cell is the controllable agent, which moves according to a random action, and the yellow cell is a fixed goal. The sequence of frames, from left to right, illustrates the movement of the agent following the action sequence: $east \rightarrow south \rightarrow east \rightarrow north$. Once the yellow cell is reached by the agent, the location of the agent and goal will be reset randomly. (b) Reconstructing the grid world into a long sequence. Each grey-shaded box contains 25 flattened grid tokens and one action token.

| Method | $l$ | Training Time (ms) | Memory Usage (%) | Error (%) |
|---|---|---|---|---|
| Mamba-2 | 208 | 25 | 13 | $15.6 \pm 2.6$ |
| | 1664 | 214 | 55 | $14.2 \pm 0.3$ |
| Mamba | 208 | 34 | 14 | $13.9 \pm 0.4$ |
| | 1664 | 299 | 52 | $14.0 \pm 0.4$ |
| GRU | 208 | 75 | 66 | $21.3 \pm 0.3$ |
| | 1664 | 628 | 68 | $34.7 \pm 25.4$ |
| Transformer | 208 | 45 | 17 | $24.7 \pm 7.4$ |
| | 1664 | - | OOM | - |

Table 2: Performance comparison of different methods in the grid world environment. Memory usage is reported as a percentage of an 8GB GPU. The error is represented as the mean $\pm$ standard deviation. The training time refers to the average duration per training step. Notably, the Transformer encounters an out-of-memory (OOM) error during training with long sequences. All experiments are conducted on a laptop. The definition of **Error (%)** is provided in Appendix A.6.

## 4 RELATED WORK

### 4.1 MODEL-BASED RL

The origin of model-based RL can be traced back to the Dyna architecture introduced by Sutton & Barto (1998), although Dyna selects actions through planning rather than learning. Notably, Sutton & Barto (1998) also highlighted the suboptimality that arises when the world model is flawed, especially as the environment improves. The concept of learning in 'imagination' was first proposed by Ha & Schmidhuber (2018), where a world model predicts the dynamics of the environment. Later, SimPLe (Kaiser et al., 2020) applied MBRL to Atari games, demonstrating improved sample effi-

ciency compared to SOTA model-free algorithms. Beginning with Hafner et al. (2019), the Dreamer series introduced a GRU-powered world model to solve a diverse range of tasks, such as MuJoCo, Atari, Minecraft, and others (Hafner et al., 2020; 2021; 2023). More recently, inspired by the success of transformers in NLP, many MBRL studies have adopted transformer architectures for their sequence models. For instance, IRIS (Micheli et al., 2023) encodes game frames as sets of tokens using VQ-VAE (Oord et al., 2017) and learns sequence dependencies with a transformer. In IRIS, the behaviour policy operates on raw images, requiring an image reconstruction during the 'imagination' process and an additional CNN-LSTM structure to extract information. TWM (Robine et al., 2023), another transformer-based world model, uses a different structure. It stacks grayscale frames and does not activate the sequence model during actual interaction phases. However, its behaviour policy only has access to limited frame history, raising questions about whether learning from tokens that already include this short-term information could be detrimental to the sequence model. STORM (Zhang et al., 2023), closely following DreamerV3, replaces the GRU with a vanilla transformer. Additionally, it incorporates a demonstration technique, populating the buffer with expert knowledge, which has shown to be particularly beneficial in the game `Freeway`.

## 4.2 Structure State space model based RL

Structured SSMs were originally introduced to tackle long-range dependency challenges, complementing the transformer architecture (Gu et al., 2022a; Gupta et al., 2022). However, Mamba and its successor, Mamba-2, have emerged as powerful alternatives, now competing directly with transformers (Gu & Dao, 2024; Dao & Gu, 2024). Deng et al. (2023) implemented an SSM-based world model, comparing it against RNN-based and transformer-based models across various prediction tasks. Despite this, while SSMs have been applied to world model-based RL (e.g., Recall to Imagine (R2I) (Samsami et al., 2024)), architectures like Mamba and Mamba-2 remain untested in this framework. Mamba has recently been applied to offline RL, either with a standard Mamba block (Lv et al., 2024) or a Mamba-attention hybrid model (Huang et al., 2024). Lu et al. (2023) proposed applying modified SSMs to meta-RL, where hidden states are manually reset at episode boundaries. Since both Mamba and Mamba-2 are input-dependent, such resets are unnecessary. Notably, R2I leverages advanced SSMs to enhance long-term memory and credit assignment in MBRL, achieving SOTA performance in memory-intensive tasks, though it exhibits slightly weaker overall performance compared to DreamerV3 (Samsami et al., 2024).

## 5 Conclusion

In conclusion, **Drama**, our proposed Mamba-based world model, addresses key challenges faced by RNN- and transformer-based world models in model-based RL. By achieving $O(n)$ memory and computational complexity, our approach enables the use of longer training sequences. Furthermore, our novel sampling method effectively mitigates suboptimality during the early stages of training, contributing to a lightweight world model (only 7 million trainable parameters) that is accessible and trainable on standard hardware. Overall, our method achieves a normalised score competitive with other SOTA RL algorithms, offering a practical and efficient alternative for model-based RL systems. Although Drama enables longer training and inference sequences, it does not demonstrate a decisive advantage that would allow it to dominate other world models on the Atari100k benchmark. An interesting direction for future work is to explore tasks where longer sequences drive superior performance in model-based RL. Additionally, it would be valuable to investigate whether Mamba can help address persistent challenges in model-based RL, such as long-horizon planning, behaviour learning, and informed exploration.

### Acknowledgments

This publication has emanated from research conducted with the financial support of Taighde Éireann - Research Ireland under Frontiers for the Future grant number 21/FFP-A/8957 and grant number 18/CRT/6223. For the purpose of Open Access, the author has applied a CC BY public copyright licence to any Author Accepted Manuscript version arising from this submission.

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

# A APPENDIX

## A.1 ATARI100K LEARNING CURVES

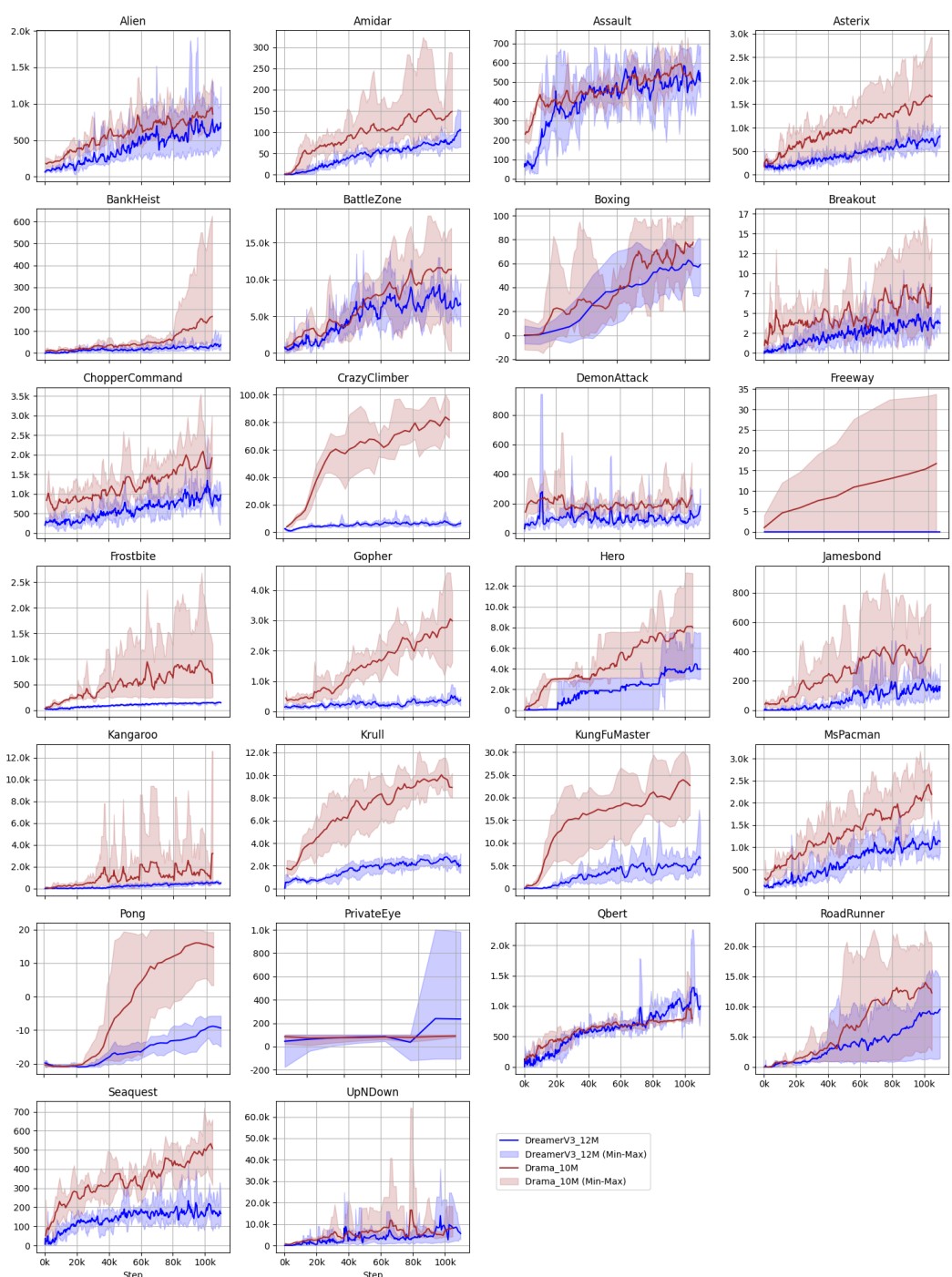

Figure 4: Atari100k Learning Curve. This figure compares the performance of DramaXS (10 million parameters) and DreamerV3XS (12 million parameters) on the Atari100k benchmark. DramaXS outperforms DreamerV3XS in most games. Exceptions include `PrivateEye` and `Qbert`, where DreamerV3XS performs better.

| Game | Random | Human | DramaXS | DreamerV3XS |
|------|--------|-------|---------|-------------|
| Alien | 228 | 7128 | **820** | 553 |
| Amidar | 6 | 1720 | **131** | 79 |
| Assault | 222 | 742 | **539** | 489 |
| Asterix | 210 | 8503 | **1632** | 669 |
| BankHeist | 14 | 753 | **137** | 27 |
| BattleZone | 2360 | 37188 | **10860** | 5347 |
| Boxing | 0 | 12 | **78** | 60 |
| Breakout | 2 | 30 | **7** | 4 |
| ChopperCommand | 811 | 7388 | **1642** | 1032 |
| CrazyClimber | 10780 | 35829 | **83931** | 7466 |
| DemonAttack | 152 | 1971 | **201** | 64 |
| Freeway | 0 | 30 | **15** | 0 |
| Frostbite | 65 | 4335 | **785** | 144 |
| Gopher | 258 | 2412 | **2757** | 287 |
| Hero | 1027 | 30826 | **7946** | 3972 |
| Jamesbond | 29 | 303 | **372** | 142 |
| Kangaroo | 52 | 3035 | **1384** | 584 |
| Krull | 1598 | 2666 | **9693** | 2720 |
| KungFuMaster | 258 | 22736 | **23920** | 4282 |
| MsPacman | 307 | 6952 | **2270** | 1063 |
| Pong | -21 | 15 | **15** | -10 |
| PrivateEye | 25 | 69571 | 90 | **207** |
| Qbert | 164 | 13455 | 796 | **983** |
| RoadRunner | 12 | 7845 | **14020** | 8556 |
| Seaquest | 68 | 42055 | **497** | 169 |
| UpNDown | 533 | 11693 | **7387** | 6511 |
| Normalised Mean (%) | 0 | 100 | 105 | 37 |
| Normalised Median (%) | 0 | 100 | 27 | 7 |

Table 3: Atari100K performance table. DramaXS achieves significantly better performance than DreamerV3XS in compact model settings within model-based reinforcement learning, highlighting the parameter efficiency of Mamba-based architectures.

## A.2 Uniform Sampling vs. DFS Learning Curves

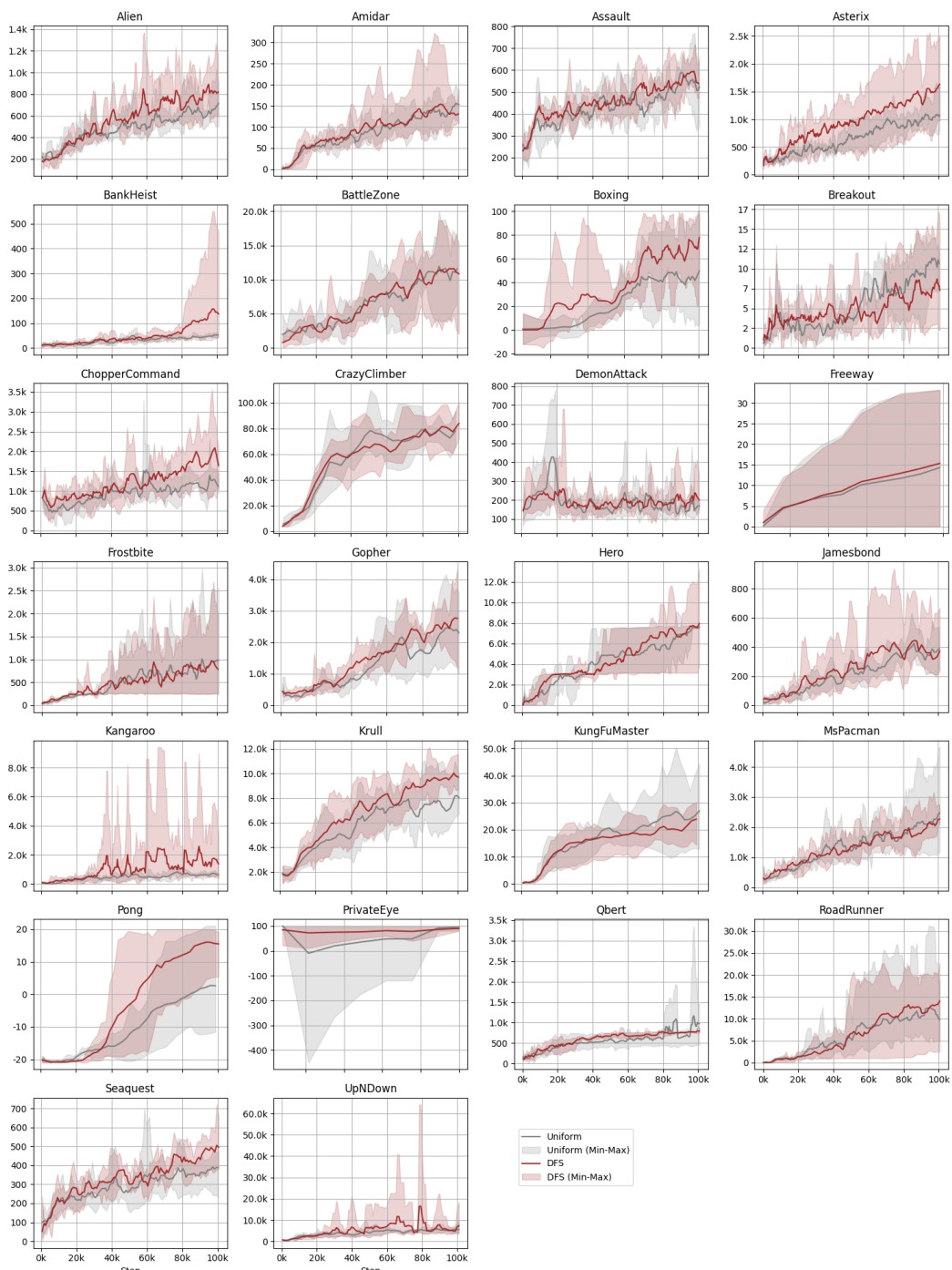

Figure 5: Uniform Sampling vs. DFS Learning Curve. DFS outperforms uniform sampling in 11 games (e.g., `Asterix`, `BankHeist`, `Krull`), underperforms in 2 games (`Breakout`, `KungFuMaster`), and matches performance in 13 games. The normalised mean score of DFS (105%) surpasses uniform sampling (80%), while the normalised median is comparable (27% vs. 28%). DFS demonstrates stronger performance in games requiring exploiting the opponents' strategy (e.g., `Pong`, `Boxing`) but struggles in environments with early-stage dynamics (Breakout).

| Game | Random | Human | DFS | Uniform |
|---|---|---|---|---|
| Alien | 228 | 7128 | **820** | 696 |
| Amidar | 6 | 1720 | 131 | **154** |
| Assault | 222 | 742 | **539** | 511 |
| Asterix | 210 | 8503 | **1632** | 1045 |
| BankHeist | 14 | 753 | **137** | 52 |
| BattleZone | 2360 | 37188 | 10860 | **10900** |
| Boxing | 0 | 12 | **78** | 49 |
| Breakout | 2 | 30 | 7 | **11** |
| ChopperCommand | 811 | 7388 | **1642** | 1083 |
| CrazyClimber | 10780 | 35829 | **83931** | 77140 |
| DemonAttack | 152 | 1971 | **201** | 151 |
| Freeway | 0 | 30 | **15** | 15 |
| Frostbite | 65 | 4335 | 785 | **975** |
| Gopher | 258 | 2412 | **2757** | 2289 |
| Hero | 1027 | 30826 | **7946** | 7564 |
| Jamesbond | 29 | 303 | **372** | 363 |
| Kangaroo | 52 | 3035 | **1384** | 620 |
| Krull | 1598 | 2666 | **9693** | 7553 |
| KungFuMaster | 258 | 22736 | 23920 | **24030** |
| MsPacman | 307 | 6952 | 2270 | **2508** |
| Pong | -21 | 15 | **15** | 3 |
| PrivateEye | 25 | 69571 | **90** | 76 |
| Qbert | 164 | 13455 | 796 | **939** |
| RoadRunner | 12 | 7845 | **14020** | 9328 |
| Seaquest | 68 | 42055 | **497** | 384 |
| UpNDown | 533 | 11693 | **7387** | 5756 |
| Normalised Mean (%) | 0 | 100 | 105 | 80 |
| Normalised Median (%) | 0 | 100 | 27 | 28 |

Table 4: The Atari100K performance table demonstrates that the Drama XS model, when paired with DFS, achieves a higher normalised mean score compared to using the uniform sampling method. This highlights the effectiveness of DFS in enhancing performance of Mamba-powered MBRL.

### A.3 MORE TRAINABLE PARAMETERS

As model-based RL agents consist of multiple trainable components, hyperparameters tuning for each part can be computationally expensive and is not the primary focus of this research. Prior work has demonstrated that increasing the neural network's size often leads to stronger performance on benchmarks (Hafner et al., 2023). In Figure 6, we demonstrate that Drama achieves overall better performance when using a more robust autoencoder and a larger SSM hidden state dimension $n$. Notably, the S model exhibits significantly improved results in games like `Breakout` and `BankHeist`, where pixel-level information plays a crucial role.

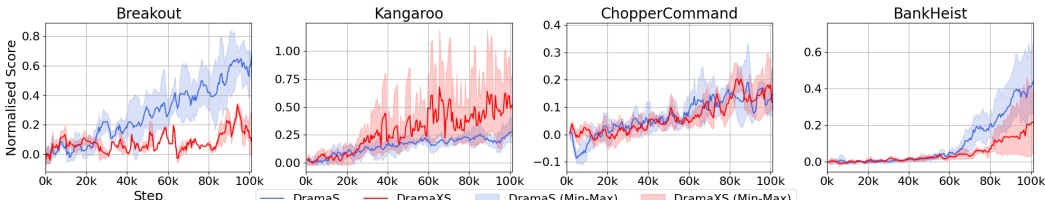

Figure 6: S model vs. XS model. We adjusted the game set to emphasise the importance of recognising small objects. The S model features a more robust autoencoder than the XS model, with additional filters and 3M more trainable parameters. In terms of performance, the S model significantly outperforms the XS model in `Breakout` and `BankHeist`. However, it underperforms in `Kangaroo` and shows comparable performance in `ChopperCommand`.

### A.4 Loss and Hyperparameters

#### A.4.1 Variational Autoencoder

The hyperparameters shown in Table 5 correspond to the default model, also referred to as XS in Figure 6. For the S model, we simply double the number of filters per layer to obtain a stronger autoencoder.

| Hyperparameter | Value |
|---|---|
| Learning rate | 4e-5 |
| Frame shape (h, w, c) | (64, 64, 3) |
| Layers | 5 |
| Filters per layer (Encoder) | (16, 32, 48, 64, 64) |
| Stride | (1, 2, 2, 2, 2) |
| Kernel | 5 |
| Weight decay | 1e-4 |
| Act | SiLU |
| Norm | Batch |

Table 5: Hyperparameters for the autoencoder.

#### A.4.2 Mamba and Mamba-2

Similar to the previous section, the values reported in Table 6 correspond to the default model. For the S model, we double the latent state dimension, thereby enabling the recurrent state to retain more task-relevant information. In the Mamba-2 model, the enhanced architecture accommodates a larger latent state dimension without a substantial increase in training time.

#### A.4.3 Reward and termination prediction heads

Both the reward and termination flag predictors take the deterministic state output from the sequence model to make their predictions. Due to the expressiveness of the temporal information extracted by the sequence model, a single fully connected layer is sufficient for accurate predictions.

| Hyperparameter | Value |
|---|---|
| Learning rate | 4e-5 |
| Hidden state dimension (d) | 512 |
| Layers | 2 |
| Latent state dimension (n) | 16 |
| Act | SiLU |
| Norm | RMS |
| Weight decay | 1e-4 |
| Dropout | 0.1 |
| Mamba-2: Head dimension (p) | 128 |

Table 6: Hyperparameters for Mamba and Mamba-2. Except the head dimension is only for Mamba-2, the other hyperparameters are shared. The head number is $512/128 = 4$.

| Hyperparameter | Value |
|---|---|
| Hidden units | 256 |
| Layers | 1 |
| Act | SiLU |
| Norm | RMS |

Table 7: Hyperparameters for reward and termination prediction heads.

The world model is optimized in an end-to-end and self-supervised manner on batches of shape $(b, l)$ drawn from the experience replay.

$$
\mathcal{L}(\omega) = \mathbb{E}\left[ \sum_{i=1}^{l} \underbrace{(O_i - \hat{O}_i)^2}_{\text{reconstruction loss}} + \mathcal{L}_{dyn}(\omega) + 0.1 * \mathcal{L}_{rep}(\omega) - \underbrace{\ln p(\hat{r}_i | d_i; \omega)}_{\text{reward prediction loss}} - \underbrace{\ln p(\hat{t}_i | d_i; \omega)}_{\text{termination prediction loss}} \right] \quad (6)
$$

where

$$
\mathcal{L}_{\text{dyn}}(\omega) = \max\left(1, \text{KL}\left[\text{sg}(p(\boldsymbol{z}_{i+1} | \mathbf{O}_{i+1}; \omega)) \,\|\, q(\hat{\boldsymbol{z}}_{i+1} | d_i; \omega)\right]\right)
$$
$$
\mathcal{L}_{rep}(\omega) = \max\left(1, \text{KL}\left[p(\boldsymbol{z}_{i+1} | \mathbf{O}_{i+1}; \omega) \,\|\, \text{sg}(q(\hat{\boldsymbol{z}}_{i+1} | d_i; \omega))\right]\right) \quad (7)
$$

and $\text{sg}(\cdot)$ represents the stop gradient operation.

### A.4.4 ACTOR CRITIC HYPERPARAMETERS

We adopt the behaviour policy learning setup from DreamerV3 (Hafner et al., 2023) for simplicity and its demonstrated strong performance, since the behaviour policy model is not central to our primary contribution.

| Hyperparameter | Value |
|---|---|
| Layers | 2 |
| Gamma | 0.985 |
| Lambda | 0.95 |
| Entropy coefficient | 3e-4 |
| Max gradient norm | 100 |
| Actor hidden units | 256 |
| Critic hidden units | 512 |
| RMS Norm | True |
| Act | SiLU |
| Batch size ($b_{img}$) | 1024 |
| Imagine context length ($l_{img}$) | 8 |

Table 8: Hyperparameters for the behaviour policy.

### A.5 PSEUDOCODE OF DRAMA

---

**Algorithm 1** Training the world model and the behaviour policy

---

**Require:** Initialize behavior policy $\pi_\theta$, world model $f_\omega$, and replay buffer $\mathcal{E}$
1: **Loop:**
2:     **Phase 1: Data Collection**
3:         Collect experience tuple $(\mathbf{O}_t, a_t, r_t, e_t)$ using $\pi_\theta$
4:         Store $(\mathbf{O}_t, a_t, r_t, e_t)$ into replay buffer $\mathcal{E}$
5:     **Phase 2: World Model Training**
6:         Sample $b$ trajectories of length $l$ from $\mathcal{E}$
7:         Update world model $f_\omega$ using sampled trajectories
8:     **Phase 3: Behaviour Model Training**
9:         Sample $b_{\text{img}}$ trajectories of length $l_{\text{img}}$ from $\mathcal{E}$
10:        Retrieve context from the first $l_{\text{img}} - 1$ experiences from the world model $f_\omega$
11:        Generate imagined rollout for $h$ steps using the last experience
12:        Train behavior policy $\pi_\theta$ with imagined rollout
13: **Repeat**

---

### A.6 THE GRID WORLD ERROR CALCULATION

The Grid World environment task requires the sequence model to capture two types of sequences. The first, referred to as the *geometric sequence*, involves reconstructing the spatial structure of the map. The environment consists of a grid surrounded by black walls, with a single agent cell and goal cell positioned, while all remaining cells are plain floor tiles. Formally, let the map $M$ be defined as a grid where $M[i, j]$ represents the cell at position $(i, j)$. The geometric sequence requires the sequence model to encode the spatial relationships such that $M[i, j]$ satisfies the constraints of walls $(W)$, floor $(F)$, agent $(A)$, and goal $(G)$, with walls forming the boundary:

$$M[i,j] = \begin{cases} W, & \text{if } (i = 0 \text{ or } i = l_g - 1) \text{ or } (j = 0 \text{ or } j = l_g - 1), \\ F, & \text{if } (i, j) \notin \{W, A, G\}, \\ A, & \text{if } (i, j) = \text{agent position}, \\ G, & \text{if } (i, j) = \text{goal position}. \end{cases}$$

The geometric error $E_g$ measures violations of the grid's structural constraints. It is defined as the number of boundary cells incorrectly classified as non-wall ($M[i, j] \neq W$ when $(i = 0 \text{ or } i = l_g - 1)$ or $(j = 0 \text{ or } j = l_g - 1)$). For interior cells, where $0 < i < l_g - 1$ and $0 < j < l_g - 1$, there must be exactly one agent and one goal, with all remaining cells being floors.

The second component, referred to as the *logic sequence*, requires predicting the agent's next position $A_t$ based on the prior action $a_{t-1}$. This prediction requires the model to retain information about the prior action, reconstruct the geometric sequence, and infer the agent's subsequent position accordingly. The logic error, $E_l$, is defined as a prediction failure, which occurs if: (1) the predicted frame contains invalid configurations (e.g., multiple agents in the interior), or (2) the predicted agent position does not match the groudtruth position in the subsequent frame.

The **Error (%)** presented in Table 2 represents the average of $E_g$ and $E_l$.

## A.7 Experiment 'Imagination' Figures

In this section, we analyze reconstructed frames generated by the 'imagination' of the sequence model to investigate potential causes of its poor performance in certain games, such as `Breakout`.

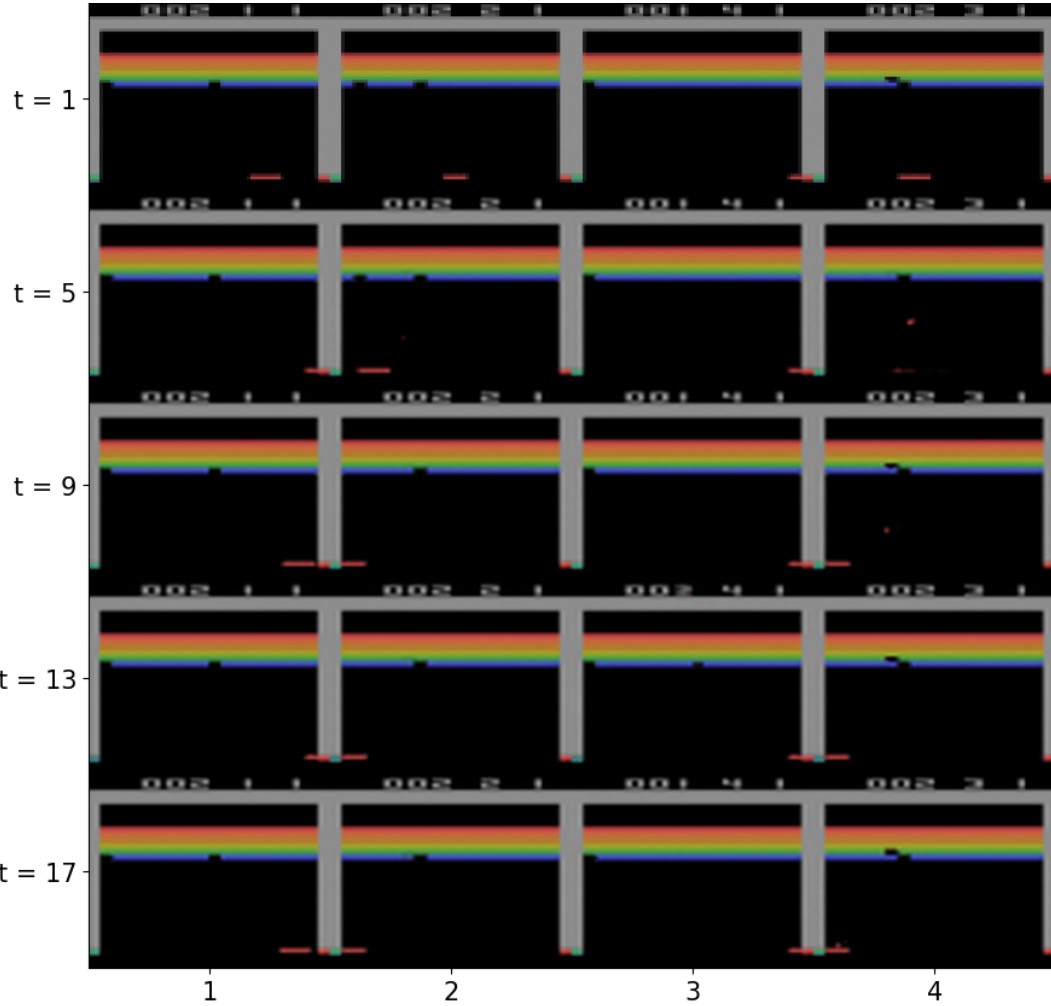

Figure 7: Drama XS model's 'imagination' in `Breakout`. The model exhibits poor performance in `Breakout`, as its autoregressive generation produces reconstructed frames that frequently omit the ball—a key visual element. This systematic omission likely undermines its ability to execute effective policies, contributing to suboptimal task performance.

The discrepancies in reconstructed frames (Figure 7, Figure 8) and the performance gains in Figure 6 collectively suggest that a more robust autoencoder enhances task performance in environments where pixel-level information is critical. This observation is further supported by the Drama XS model's strong performance in `Pong` (Figure 9), a game sharing core mechanics with `Breakout` (e.g., paddles and balls) but with reduced visual complexity due to the absence of multicolored bricks. While systematic analysis is warranted to validate this hypothesis, these results indicate that refining the autoencoder may serve as a critical first step in alleviating performance limitations in visually demanding tasks.

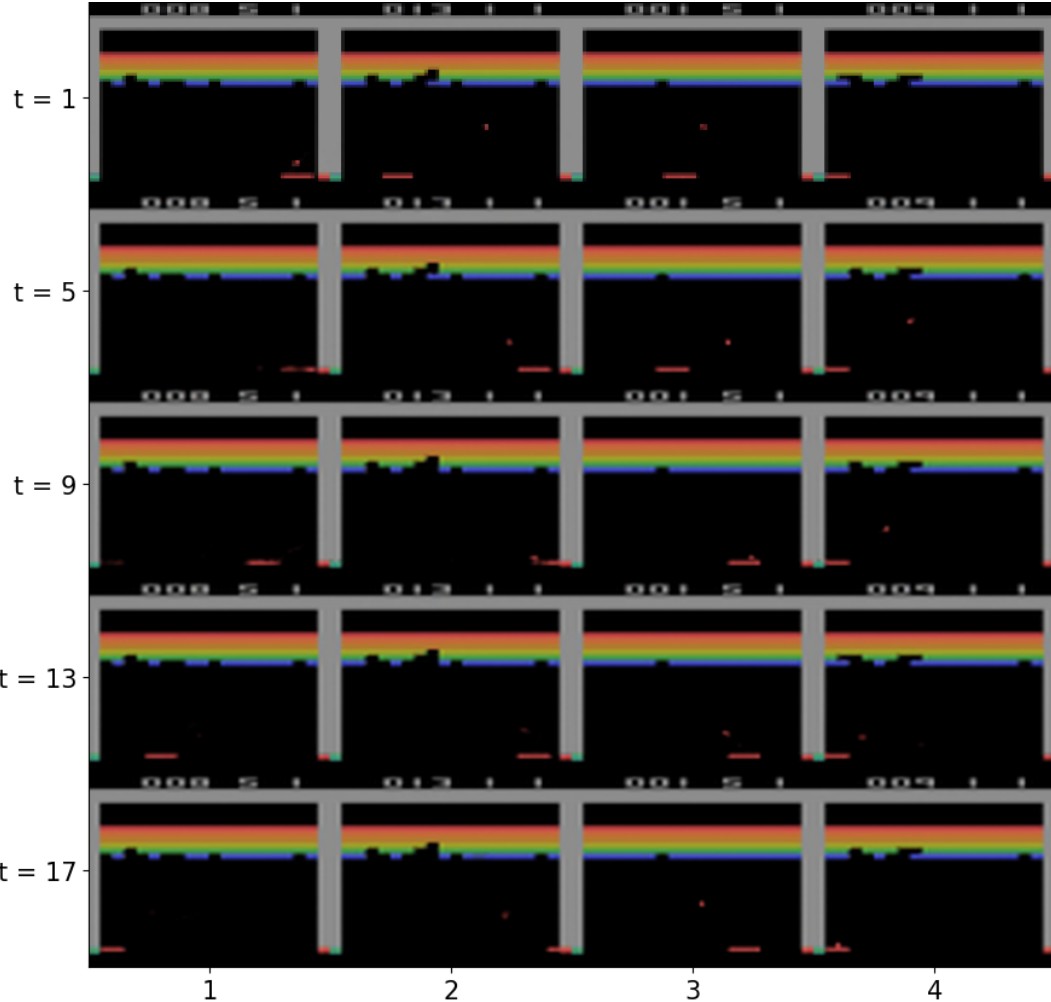

Figure 8: Drama S model's 'imagination' in `Breakout`. The Drama S model exhibits significant improvements over the XS variant, with the ball—a critical game element—consistently reconstructed in the majority of autoregressive frames. This enhancement suggests a stronger capacity to encode pixel-level details, aligning with its superior task performance.

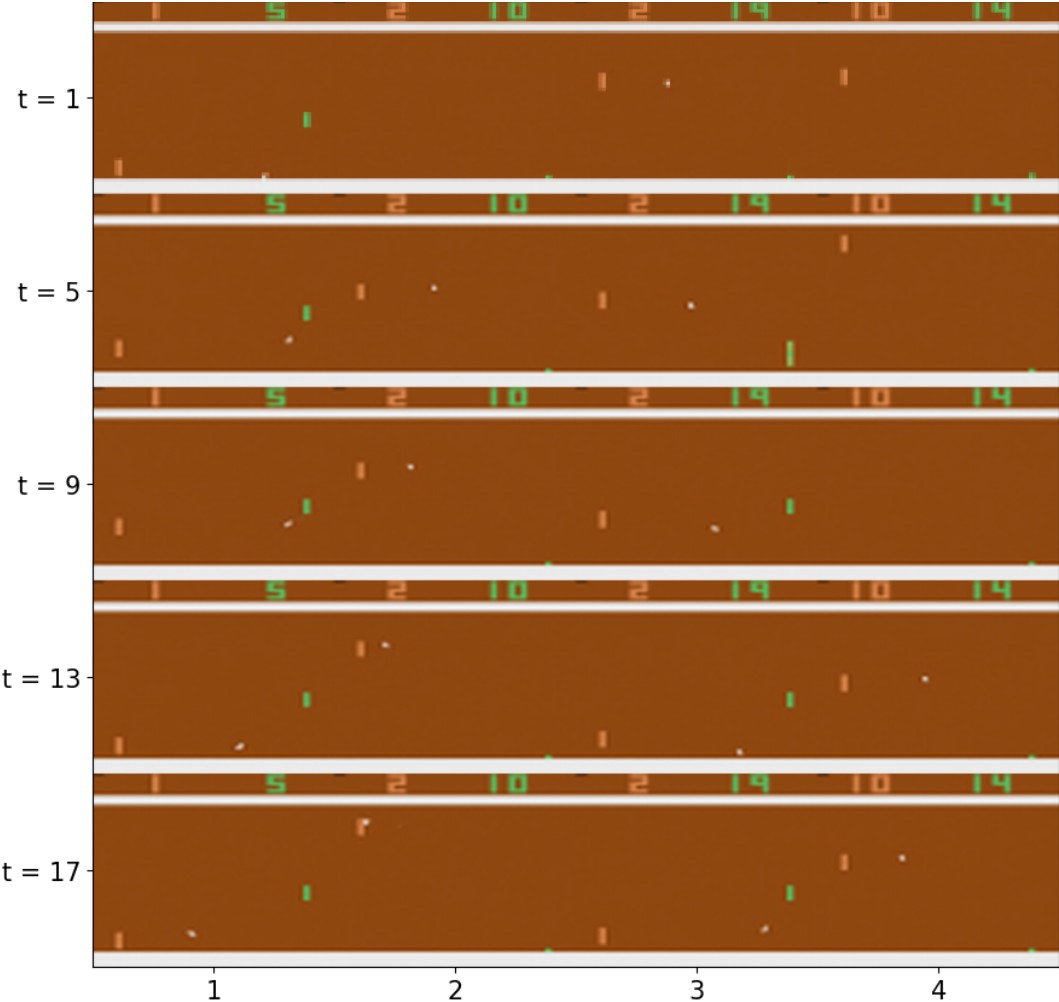

Figure 9: Drama XS model's 'imagination' in `Pong`. The Drama XS model exhibits strong performance in `Pong`, contrasting sharply with its suboptimal results in `Breakout`. While both games share core mechanics (e.g., paddles and balls), `Pong`'s absence of multicolored bricks reduces visual complexity, thereby lowering demands on the model's frame-encoding capacity. Consequently, the ball—a critical element—is consistently reconstructed in the majority of autoregressive frames, supporting effective policy execution.

## A.8 Wall-Clock Time Comparison of Sequence Models in MBRL

As illustrated in Figure 10, we compare the wall-clock time efficiency of sequence models in the Atari100k MBRL task. The results demonstrate that Mamba and Mamba-2 outperform the Transformer architecture during the imagination phase for all tested sequence lengths. While Mamba-2 exhibits a marginal computational overhead compared to Mamba and the Transformer for shorter training sequences, it achieves superior efficiency for longer sequences, making it particularly advantageous for tasks demanding long-range temporal modelling. All models were evaluated under identical experimental conditions, with comparable parameter sizes and training configurations.

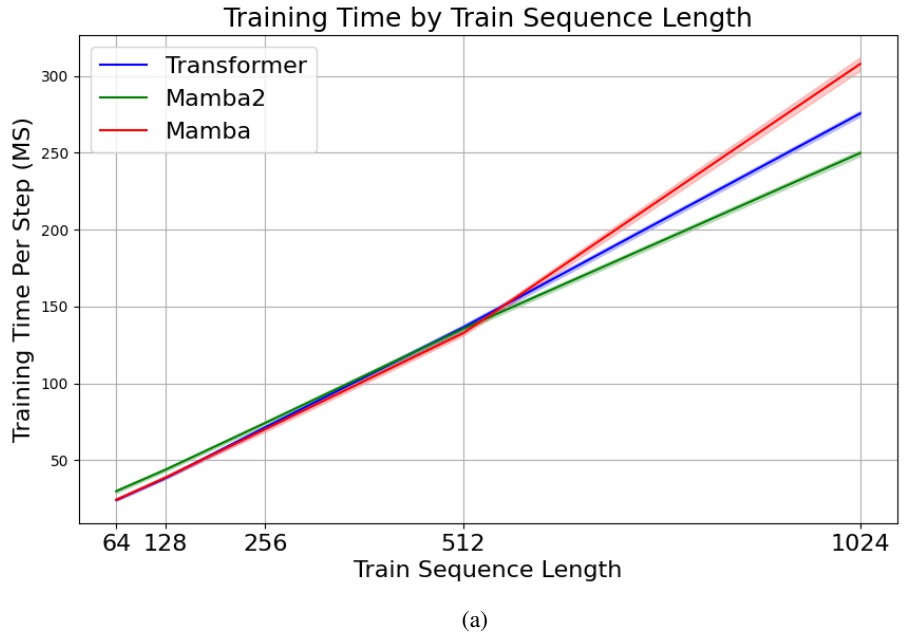

(a)

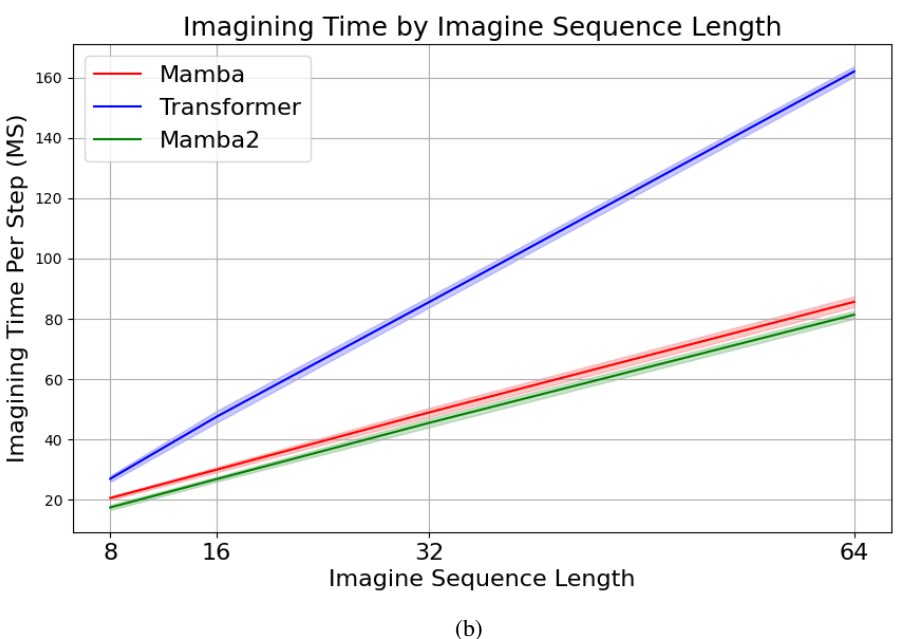

(b)

Figure 10: Wall-clock time comparison of sequence models in MBRL. Experiments were conducted on a consumer-grade laptop with an NVIDIA RTX 2000 Ada Mobile GPU, ensuring practical relevance to resource-constrained settings. Notably, the Transformer model leveraged a key-value (KV) cache to optimise inference speed. Results demonstrate that Mamba-2 achieves superior efficiency for longer sequences in both training and 'imagination' phases. However, it incurs a slight computational overhead compared to the Transformer and Mamba during training at shorter sequence lengths.

