# OpenReview forum: "Drama: Mamba-Enabled Model-Based Reinforcement Learning Is Sample and Parameter Efficient"
_ICLR.cc/2025/Conference — ICLR 2025 Poster_

### Official Review · Reviewer_mCEd · 2024-10-27

**Soundness:** 3
**Presentation:** 3
**Contribution:** 3
**Rating:** 6
**Confidence:** 4

**Summary:**

The authors propose a MAMBA-based world model architecture, as opposed to previous transformer and RSSM based ones. They also compare between MAMBA-1 and MAMBA-2 for world models. Finally, they evaluate and ablate their technique on the Atari100k benchmark.

**Strengths:**

- The combination of MAMBA and Sequence-based world models is novel
- The authors demonstrate comparable results with a smaller model size

**Weaknesses:**

- Some results are unclear. For example, could you go into details on why the model is not doing well on the Breakout game? What is it producing? Some figures from the game can be also useful--not just this one, but the ones in which model does well too.
- It is not clear why imagination context length is kept to 8. I would suggest providing experiments, both involving time-complexity and performance, for different imagination context length. Also, explaining why this is done would be useful.

**Questions:**

- What is the value of h?
- "A key difference between Mamba-based and transformer-based world models in the ‘imagination’ process is that Mamba updates inference parameters independent of sequence length."--can you explain it more?

---

> ### Author Response · Authors · 2024-11-27
>
> > Some results are unclear. For example, could you go into details on why the model is not doing well on the Breakout game? What is it producing? Some figures from the game can be also useful--not just this one, but the ones in which model does well too.
>
> We assume the reviewer's question arises because *Drama* performs well in the game *Pong* but not in *Breakout*, despite the two games sharing some similar features. The reason is that *Breakout* is more visually complex due to its colorful bricks, causing the encoder to fail in effectively encoding the ball. As requested, we have added a subsection in the appendix to explain this in detail with the experiment figures. Please refer to Section A.7 in the revised version.
>
>
>
> > It is not clear why imagination context length is kept to 8. I would suggest providing experiments, both involving time-complexity and performance, for different imagination context length. Also, explaining why this is done would be useful.
>
> MBRL involves numerous hyperparameters, and conducting a comprehensive hyperparameter search demands substantial computational resources and time. Consequently, we adopt the hyperparameters established in prior research studies. This approach is commonly employed in the literature.
> For example, several studies such as TWM (Robine et al., 2023), STORM (Zhang et al., 2023), Hieros (Mattes et al., 2024), and DreamerV3 (Hafner et al., 2024) utilise the same imagination horizon, originally introduced in DreamerV1. Specifically, the imagination context length in this work aligns with the hyperparameter used in STORM.
>
> **Reference:**
>
> **Zhang, Weipu, Gang Wang, Jian Sun, Yetian Yuan, and Gao Huang.**
>   *“STORM: Efficient Stochastic Transformer Based World Models for Reinforcement Learning.”*
>   In *Thirty-Seventh Conference on Neural Information Processing Systems*, 2023.
>
> **Robine, Jan, Marc Höftmann, Tobias Uelwer, and Stefan Harmeling.**
>   *“Transformer-Based World Models Are Happy With 100k Interactions.”*
>   In *International Conference on Learning Representations*, 2023.
>
> **Mattes, Paul, Rainer Schlosser, and Ralf Herbrich.**
>   *“Hieros: Hierarchical Imagination on Structured State Space Sequence World Models.”*
>   In *Forty-First International Conference on Machine Learning*, 2024.
>
> **Hafner, Danijar, Jurgis Pasukonis, Jimmy Ba, and Timothy Lillicrap.**
>   *“Mastering Diverse Domains through World Models.”*
>   *arXiv*, 17 April 2024.
>
> > What is the value of h?
>
> We assume that $h$ refers to the dimension of the hidden state in Mamba. Specifically, $h$ is 16 for the XS model and 32 for the S model.
>
> > "A key difference between Mamba-based and transformer-based world models in the ‘imagination’ process is that Mamba updates inference parameters independent of sequence length."--can you explain it more?
>
> Yes, during imagination, Mamba (both versions 1 and 2) utilises a hidden state to summarize all past information. The hidden state has a fixed dimensionality (16 or 32 in the examples above), the model updates the hidden state at time step $t$ without reprocessing the past token $x_{t-1}$. Consequently, the inference time scales linearly with the sequence length.
> This scalability is illustrated in Section A.8, Figure 10 (A) of the revised version, showing that Mamba-based world models are faster for "imagination" compared to Transformer-based world models.

---

> > ### Author Response · Authors · 2024-11-29
> > **Kind Reminder to Activate the Discussion Before December 2nd**
> >
> > Dear reviewer mCEd,
> >
> > I wanted to kindly remind you that the deadline to respond to reviews and participate in the discussion is Monday, December 2nd AoE. Since the weekend is approaching, we understand you might have limited time afterward to engage.
> >
> > We value your insights and are eager to activate a productive discussion before the deadline. If possible, we would greatly appreciate it if you could share your thoughts soon. Thank you for your time and contributions to the review process.
> >
> > Best regards,
> >
> > Authors

---

> > > ### Comment · Reviewer_mCEd · 2024-12-01
> > >
> > > Thank you for your response. I have updated the score to 6.

---

> > > > ### Author Response · Authors · 2024-12-01
> > > >
> > > > Thank you for your questions and insights.

---

### Official Review · Reviewer_mwc8 · 2024-10-31

**Soundness:** 3
**Presentation:** 3
**Contribution:** 3
**Rating:** 8
**Confidence:** 3

**Summary:**

This paper presents a new world model for deep reinforcement learning that is based on the Mamba-2 architecture. In addition, the authors introduce dynamic frequency-based sampling, which leads to more accurate imaginations of the world model. The approach achieves competitive performance on the Atari 100k benchmark while maintaining a lightweight architecture with only 7 million trainable parameters.

**Strengths:**

- [S1] The paper combines established methods in a novel way, effectively addressing an existing gap in world model research.
- [S2] The proposed model is computationally efficient, requiring only 7 million trainable parameters, making it accessible.

**Weaknesses:**

- [W1] The extent to which the Mamba architecture contributes to the model's performance remains unclear. Specifically, it is unclear how DFS impacts scores across all games. Extending ablation study 3.2.1 to cover more games, or conducting a new study that replaces Mamba with an RNN or transformer, would clarify these contributions.
- [W2] While the paper emphasizes Mamba's computational efficiency, there is a lack of exact wall-clock training and inference times. The abstract claims the model can be trained on a standard laptop, so providing specific runtime metrics would substantiate this claim.
- [W3] Several presentation aspects should be improved:
  - The paragraph in lines 99–105 shifts from general statements about model-based RL to specific details about the paper's world model. This transition could lead readers to infer that every world model relies on a variational autoencoder or linear heads, which isn't necessarily the case. Additionally, other model-based RL methods exist that don't utilize world models, such as those using lookahead search.
  - Many figures contain small text that is difficult to read without zooming.
  - Minor suggestions:
    - Highlight the highest scores in Table 1 for easy reference.
    - Correct notations for all \hat{} terms (e.g., \hat{r}_t instead of \hat{r_t}).
    - Address minor notation errors: e.g., incorrect $A$ on lines 148, 154, 169 and incorrect $T$ on line 177
    - Variables on line 218 should be in math mode.
    - Consider changing "auto-generative" to "autoregressive" on line 238?
    - Revise line 264 to read "tracks the number of *times* the transition has been used."
- [W4] To strengthen the soundness of findings, additional evaluations on alternative benchmarks, such as the DeepMind Control Suite, would be valuable. That said, I understand this may be challenging to realize.

I would consider raising my scores if these issues were addressed.

**Questions:**

- [Q1] Could the decoder operate based on the output of Mamba-2, such that $d$ rather than $z$ serves as the input?

---

> ### Author Response · Authors · 2024-11-26
>
> > [W1] The extent to which the Mamba architecture contributes to the model's performance remains unclear. Specifically, it is unclear how DFS impacts scores across all games. Extending ablation study 3.2.1 to cover more games, or conducting a new study that replaces Mamba with an RNN or transformer, would clarify these contributions.
>
> As noted above, to fully address this concern, we conducted a study comparing uniform sampling and DFS in DRAMA across all games. The results demonstrate the effectiveness of DFS, which outperformed uniform sampling in 11 games, underperformed in 2 games, and tied in 13 games.
>
> > [W2] While the paper emphasizes Mamba's computational efficiency, there is a lack of exact wall-clock training and inference times. The abstract claims the model can be trained on a standard laptop, so providing specific runtime metrics would substantiate this claim.
>
> We did not explicitly state that Drama is computationally efficient in the paper, as MBRL is naturally more computationally complex than model-free RL due to the involvement of the world model. However, we mentioned that SSMs achieve $O(n)$ memory and computational complexity, where $n$ represents the sequence length. We interpret the reviewer's question as a request for proof of this claim.
>
> Since we are using a shared server for training, it is difficult to ensure a clean and consistent environment to test the wall-clock training time. To address this, we evaluated the training time solely for the model using the grid world environment and have included the results in the revised version. Please refer to *Table 2* for the result and section 3.2.3 for the detail in the revised version.
>
>  > The paragraph in lines 99–105 shifts from general statements about model-based RL to specific details about the paper's world model. This transition could lead readers to infer that every world model relies on a variational autoencoder or linear heads, which isn't necessarily the case. Additionally, other model-based RL methods exist that don't utilize world models, such as those using lookahead search.
>
>  We have revised the text for clarity and included references to additional approaches.
>
> "There are various approaches to obtaining a world model, including Monte Carlo tree search (Schrittwieser et al., 2020), offline imitation learning (DeMoss et al., 2023) and latent dynamics models (Hafner et al., 2019). In this work, we focus on learning a world model $ f(O_t, a_t; \omega)$ from actual experiences to capture the dynamics of the POMDP in a latent space."
>
> References:
>
> Schrittwieser, Julian, Ioannis Antonoglou, Thomas Hubert, Karen Simonyan, Laurent Sifre, Simon Schmitt, Arthur Guez, et al. **“Mastering Atari, Go, Chess and Shogi by Planning with a Learned Model.”** *Nature* 588, no. 7839 (24 December 2020): 604–609. [https://doi.org/10.1038/s41586-020-03051-4](https://doi.org/10.1038/s41586-020-03051-4).
>
> Hafner, Danijar, Timothy Lillicrap, Ian Fischer, Ruben Villegas, David Ha, Honglak Lee, and James Davidson. **“Learning Latent Dynamics for Planning from Pixels.”** In *Proceedings of the 36th International Conference on Machine Learning*, 97:2555–2565. PMLR, 2019.
>
> DeMoss, Branton, Paul Duckworth, Nick Hawes, and Ingmar Posner. **“DITTO: Offline Imitation Learning with World Models.”** *arXiv*, 6 February 2023. [http://arxiv.org/abs/2302.03086](http://arxiv.org/abs/2302.03086).

---

> > ### Author Response · Authors · 2024-11-26
> >
> > > Many figures contain small text that is difficult to read without zooming.
> >
> > We have updated the figures to use a larger font size.
> >
> > > Minor suggestions
> >
> > We have incorporated the suggestions, except for the point regarding "Address minor notation errors," as we are unsure which notation errors the reviewer is referring to. If the font differences were the issue, the varying font styles align with the ICLR 2025 author's guidelines, where tensors, matrices, and vectors are represented in distinct font styles. If this does not address the notation errors the reviewer had in mind, please feel free to clarify. Thank you.
> >
> > > To strengthen the soundness of findings, additional evaluations on alternative benchmarks, such as the DeepMind Control Suite, would be valuable. That said, I understand this may be challenging to realize.
> >
> > We appreciate the reviewer’s suggestion to extend our experiments to additional benchmarks, such as the DeepMind Control Suite. However, due to limited computational resources, it is not feasible for us to conduct these additional experiments within the constraints of this review process.
> >
> > We would also like to emphasize that prior works in this domain have predominantly utilized Atari100k as the sole benchmark, which is widely recognized as a standard for evaluating algorithms under similar conditions [Kaiser et al., 2020; Micheli et al., 2023; Robine et al., 2023; Zhang et al., 2023]. We believe our current evaluation on Atari100k provides a robust and fair comparison to prior work.
> >
> > References:
> >
> > Kaiser, Lukasz, Mohammad Babaeizadeh, Piotr Milos, Blazej Osinski, Roy H. Campbell, Konrad Czechowski, Dumitru Erhan, et al. **“Model-Based Reinforcement Learning for Atari.”** In *International Conference on Learning Representations*, 2020.
> >
> > Micheli, Vincent, Eloi Alonso, and François Fleuret. **“Transformers Are Sample-Efficient World Models.”** In *International Conference on Learning Representations*, 2023.
> >
> > Robine, Jan, Marc Höftmann, Tobias Uelwer, and Stefan Harmeling. **“Transformer-Based World Models Are Happy With 100k Interactions.”** In *International Conference on Learning Representations*, 2023.
> >
> > Zhang, Weipu, Gang Wang, Jian Sun, Yetian Yuan, and Gao Huang. **“STORM: Efficient Stochastic Transformer Based World Models for Reinforcement Learning.”** In *Thirty-Seventh Conference on Neural Information Processing Systems*, 2023.
> >
> > > [Q1] Could the decoder operate based on the output of Mamba-2, such that $d$
> >  rather $z$ than serves as the input?
> >
> > We did not test using $d_t$ as the input to the decoder, but we did test using $\hat{z}_{t+1}$, which is generated from $d_t$, as input during the early stages of development. The results were consistent with those reported by Zhang et al. (2023) in Section 5.1 of their paper, where they describe the "decoder at rear" setup. As noted in their findings, this approach leads to poor performance. Therefore, we chose not to include these results in our paper.
> >
> > Since $d_t$ includes context information that may not be necessary for reconstructing the current observation, we hypothesize that the results would likely be similar, leading to degraded performance. However, we do not have experimental results to confirm this hypothesis.
> >
> > Reference:
> >
> > Zhang, Weipu, Gang Wang, Jian Sun, Yetian Yuan, and Gao Huang. **“STORM: Efficient Stochastic Transformer Based World Models for Reinforcement Learning.”** In *Thirty-Seventh Conference on Neural Information Processing Systems*, 2023.

---

> > > ### Comment · Reviewer_mwc8 · 2024-11-27
> > >
> > > Thank you for your detailed responses to my comments and my question and for revising the manuscript accordingly.
> > > That said, I still have some concerns, which I've outlined further in my comments.
> > >
> > > > We did not explicitly state that Drama is computationally efficient in the paper, as MBRL is naturally more computationally complex than model-free RL due to the involvement of the world model. However, we mentioned that SSMs achieve O(n) memory and computational complexity, where n represents the sequence length.
> > >
> > > Thank you for your response and the updates to the manuscript, including the additional evaluation of training time in the grid world environment. While I understand that wall-clock comparisons can be challenging, I believe there is still an opportunity to clarify and substantiate the computational efficiency claims.
> > >
> > > While the paper may not explicitly state that Mamba is computationally efficient, the abstract strongly implies it. For example, the first paragraph raises concerns about the computational cost of learning robust world models, while the second paragraph describes Mamba as addressing these challenges, using only 7 million parameters, and being trainable on an off-the-shelf laptop. These statements collectively create an expectation that Mamba offers computational advantages over comparable approaches.
> > >
> > > To strengthen the paper and align it with these expectations, I suggest including a comparison of training times or computational resource requirements with other world models, such as Dreamer or IRIS, particularly on a more complex benchmark like Atari 100k. Even an approximate order-of-magnitude comparison would provide valuable context for readers and give a clearer sense of how Mamba's computational properties translate to practical scenarios.
> > >
> > > This additional insight would further highlight the accessibility and efficiency of the method, which I believe are key selling points of the work.
> > >
> > > > We have incorporated the suggestions, except for the point regarding "Address minor notation errors," as we are unsure which notation errors the reviewer is referring to. If the font differences were the issue, the varying font styles align with the ICLR 2025 author's guidelines, where tensors, matrices, and vectors are represented in distinct font styles. If this does not address the notation errors the reviewer had in mind, please feel free to clarify. Thank you.
> > >
> > > My apologies for any confusion caused by my previous comment. I believe there are still a couple of instances where the notation could be improved for consistency and clarity:
> > > - In lines 146 and 148, the matrix $A$ should be represented in the same font style. Since $A$ is introduced as a matrix, I recommend updating the notation in line 148 to reflect this.
> > > - In line 177, the $T$ in $\mathbb{R}^{(T,T)}$ should align with the regular $T$ introduced in line 154 for consistency.
> > >
> > > > However, due to limited computational resources, it is not feasible for us to conduct these additional experiments within the constraints of this review process.
> > >
> > > I understand that extending the experiments to other benchmarks is difficult due to computational constraints, and I acknowledge the standard practice of evaluating on the Atari 100k benchmark.
> > >
> > > Based on the revisions and clarifications provided, I have increased my scores slightly.

---

> > > > ### Author Response · Authors · 2024-11-27
> > > >
> > > > > To strengthen the paper and align it with these expectations, I suggest including a comparison of training times or computational resource requirements with other world models, such as Dreamer or IRIS, particularly on a more complex benchmark like Atari 100k. Even an approximate order-of-magnitude comparison would provide valuable context for readers and give a clearer sense of how Mamba's computational properties translate to practical scenarios.
> > > >
> > > > We agree with the reviewer that adding a section comparing the training time and "imagination" time across different dynamics models in MBRL would strengthen the paper.
> > > > In the revised version, we have added Section A.8 in the appendix. This section includes wall-clock (on a laptop) comparisons between the Mamba-based world model and the Transformer-based world model. The results demonstrate that Mamba-based world models (both Mamba-1 and Mamba-2) are faster in "imagination" for the tested sequence lengths. While Mamba-2 is slightly slower during training with short sequence lengths, it catches up as the training sequence length increases.
> > > >
> > > > We did not test DreamerV3 because it is implemented in JAX rather than PyTorch, making the wall-clock comparisons inconsistent. However, we used a Transformer model similar to STORM, and in the work by Zhang et al. (2023), they reported more efficient training performance compared to DreamerV3.
> > > >
> > > >
> > > >
> > > > > Notation typos
> > > >
> > > > Thank you for pointing out the notation typos. We have corrected the notation fonts as suggested.

---

> > > > > ### Comment · Reviewer_mwc8 · 2024-11-28
> > > > >
> > > > > Thank you for addressing my concerns. I believe the paper is now in a good state and would make a valuable contribution to the field. Therefore, I have increased my scores.

---

> > > > > > ### Author Response · Authors · 2024-11-28
> > > > > >
> > > > > > Thank you for your advice and for recognising Drama's contribution to the field.

---

### Official Review · Reviewer_hT9b · 2024-11-03

**Soundness:** 3
**Presentation:** 2
**Contribution:** 2
**Rating:** 6
**Confidence:** 4

**Summary:**

This paper proposes using State-Space Models (SSMs) for learning World Models. Specifically, the architecture comprises an encoder to get discrete latents, a SSM module (Mamba-2) to estimate the dynamics which is used to predict the latent embedding of observation, reward value and termination flag. This world model is used to train a policy by imagination. The paper also uses a method to sample transitions from the replay buffer based on the number of times the transition is used to update WM and policy. Experiments conducted on Atari100K show that the proposed method Drama attains similar performance to IRIS and TWM with a much smaller model in terms of parameters.

**Strengths:**

- The proposed idea of using SSM for WMs is interesting as they provide crucial benefits over training with Transformers and RNNs.
- The proposed method achieves good performance with significantly fewer parameters (7M) when compared with baselines.

**Weaknesses:**

- It is hard to articulate where the performance gains are coming from. Section 3.2.1 discusses that DFS provides an advantage over uniform sampling. Since DFS is agnostic to most baselines, it is important to see a comparison of either Drama with Uniform Sampling or baselines with DFS sampling to understand if the architecture is helping or the sampling.
- The paper is not well written and it is hard to understand the details and motivation behind the design choices. Questions 1-5 below expand on this. The paper can use a pseudocode to describe the behavior learning part.

**Questions:**

1. At line 222, it is not clear what targets mean.
2. Section 2.3 is not described well. At line 240, when the ‘b’ starting points are sampled of length $l_{img}$, is it just picking random samples along $b_{img}$ sequences? Since the batch sampled from replay buffer is of length $l_{img}$, I am unsure what this additional sampling is doing? Why not just sample $b$ trajectories from the buffer?
3. What is $h_t$ in behavior policy learning? The deterministic variable is defined as $d_t$ in Eq 5.
4. While training Dreamer, the method uses the whole sequence of ($b_{img}, l_{img}$) to compute a good hidden state and uses all sampled to generate trajectories in the future. I am curious to know why only the last hidden state $l_{img}$ is used for learning the policy and not the whole sequence like Dreamer? (describe around line 243).
5. The terminal flag is defined as $e_t$ at line 93, whereas it is $t_i$ in description of Figure 1. Also, the description of Fig 1 uses $i$ for indexing the time and the Section 2 starts with $t$ as time index. The description of Fig 1 should match the notation in the main text.
6. Was any experiment conducted to see if the behavior model underestimates rewards especially with limited data?
7. For the ablation presented in Sec 3.2.2, were both variants Mamba-1 and Mamba-2 based WM trained with DFS?
8. How does the proposed method compare with R2I [1]. Since they propose using SSMs, should it be included as another baseline?
9. [Typo] Employs is written twice at line 132.

#### References
[1] Samsami et al., Mastering Memory Tasks with World Models, ICLR’24.

---

> ### Author Response · Authors · 2024-11-26
>
> > Since DFS is agnostic to most baselines, it is important to see a comparison of either Drama with Uniform Sampling or baselines with DFS sampling to understand if the architecture is helping or the sampling.
>
> As noted above, to fully address this concern, we conducted a study comparing uniform sampling and DFS in Drama across all games. The results demonstrate the effectiveness of DFS, which outperformed uniform sampling in 11 games, underperformed in 2 games, and tied in 13 games.
>
>
> > At line 222, it is not clear what targets mean.
>
> It has been addressed. We expanded the writing to make it clearer.
> Updated lines in the revised version: 221-225 (Some latex formular is not supported here so we can't copy the revised sentence to here.)
>
> > Section 2.3 is not described well.
>
> We have updated the text to clarify that we sample $b_{img}$ trajectories, each of length $l_{img}$.
>
> "The behaviour policy is trained within the `imagination', an autoregressive process driven by the dynamics model. Specifically, a batch of $\displaystyle b_{img}$ trajectories each of length $l_{img}$ is sampled from the replay buffer. "
>
> > What is $h_t$ in behavior policy learning? The deterministic variable is defined as $d_t$ in Eq 5.
>
> Thank you for pointing this out. It was a typo, which has now been corrected.
>
> > While training Dreamer, the method uses the whole sequence of $(b_{img}, l_{img})$ to compute a good hidden state and uses all sampled to generate trajectories in the future. I am curious to know why only the last hidden state $l_{img}$ is used for learning the policy and not the whole sequence like Dreamer? (describe around line 243).
>
> Dreamer samples one batch of trajectories, typically in the shape (64, 16). These samples are then used to generate rollouts of length 15, resulting in a total of (1024, 16) samples to train the behavior model. However, since some starting points are consecutive, this can lead to overlapping and correlated imagined trajectories. To address this, we resample $b_{img} = 1024$ trajectories directly from the buffer to increase the diversity of the training samples. To ensure the same batch size of rollout, we only use the last hidden state $l_{img}$ to generate rollout with the horizon $h=15$, while while the sequence preceding $l_{img}$ is used solely to bootstrap the hidden state of Mamba2.
>
> > The terminal flag is defined as $e_t$ at line 93 ...
>
> Thank you for pointing this out. We have updated the notation in the figure and legend to align with the main text.
>
> > Was any experiment conducted to see if the behavior model underestimates rewards especially with limited data?
>
> We did not conduct any experiments specifically to examine this phenomenon. However, it is well explained with a simple yet convincing example in Chapter 8.3 of [Sutton & Barto, 2018]. In MBRL, the behavior model may overestimate rewards in states where the world model is underfitting—a phenomenon known as the *model exploitation problem*. This issue remains a significant challenge in model-based RL.
>
> Theoretically, DFS can help mitigate both problems. It increases the likelihood of sampling fresh trajectories to train the world model and decreases the likelihood of sampling trajectories where the world model is underfitting. However, DFS does not fully resolve these issues.
>
> **Reference**:
> Sutton, Richard S., and Andrew G. Barto. *Reinforcement Learning: An Introduction*. MIT Press, 2018.
>
> > For the ablation presented in Sec 3.2.2, were both variants Mamba-1 and Mamba-2 based WM trained with DFS?
>
> Yes, We stated it in the main text line 398. We changed Figure 2's legend to further clearify it.
>
> > How does the proposed method compare with R2I [1]. Since they propose using SSMs, should it be included as another baseline?
>
> Thank you for highlighting this reference; we will include it in our related work. As mentioned earlier, we currently lack sufficient computational resources to evaluate DRAMA on the POPgym and Maze baselines. However, in the Atari100k benchmark, DRAMA outperforms R2I. We believe Atari100k provides a reasonable benchmark to demonstrate the model's capabilities. That said, we agree that including POPgym and Maze baselines would enhance the evaluation, and we will consider these benchmarks in future work.

---

> > ### Author Response · Authors · 2024-11-29
> > **Kind Reminder to Activate the Discussion Before December 2nd**
> >
> > Dear Reviewer hT9b,
> >
> > I wanted to kindly remind you that the deadline to respond to reviews and participate in the discussion is Monday, December 2nd AoE. Since the weekend is approaching, we understand you might have limited time afterward to engage.
> >
> > We value your insights and are eager to activate a productive discussion before the deadline. If possible, we would greatly appreciate it if you could share your thoughts soon. Thank you for your time and contributions to the review process.
> >
> > Best regards,
> >
> > Authors

---

> > > ### Comment · Reviewer_hT9b · 2024-12-02
> > >
> > > Happy to see many of my concerns being addressed in the rebuttal. I am updating my score. It is interesting to see Drama works better than R2I on Atari-100K.
> > >
> > > However, I am still not fully convinced by the response of underestimation of rewards. Currently, DFS increases the likelihood of new trajectories, but it is unsure what happens when the new data is quite similar to the content in the replay buffer-- which is why exploration bonuses are preferred that provide an incentive to visit uncertain states more often.

---

> > > > ### Author Response · Authors · 2024-12-02
> > > >
> > > > Thank you for the response. I’m glad to hear that many concerns have been addressed.
> > > >
> > > > >  exploration bonuses are preferred that provide an incentive to visit uncertain states more often
> > > >
> > > > This is a very interesting insight. I am very interested in intrinsic reward-based RL. I believe that agents interacting with the real world and leveraging exploration bonuses—such as state entropies, prediction errors, etc.—can generate more diverse training data for world models. This creates an intriguing direction for research: using an intrinsic agent, denoted as $\pi_\phi$, to collect trajectories for training the world model, while simultaneously training a behavior model, $\pi_\theta$, to maximise the task reward.
> > > >
> > > > ### Challenges
> > > > However, this approach introduces some key challenges:
> > > >
> > > > 1. **Mismatch in State Distributions**:
> > > >    The state distribution induced by $\pi_\phi$, denoted $Pr_\phi(S)$, differs from the distribution induced by $\pi_\theta$, $Pr_\theta(S)$. Since $\pi_\theta$ is trained under the "imagination" of the world model—which itself is trained on data collected by $\pi_\phi$—there is a risk of $\pi_\theta$ learning from trajectories it would never naturally encounter. For instance, in a game like *Pong*, $\pi_\phi$ might explore a rare state such as losing 0:18, which a well-trained $\pi_\theta$ would almost never reach in actual gameplay.
> > > >
> > > > 2. **Training on a Broader Distributions**:
> > > >    Training $\pi_\theta$ with such a broad distribution could ultimately enhance its robustness, but it would likely require significantly more samples to achieve convergence. DFS is helpful when the same agent is used for both training the world model and collecting data in the real game. However, this doesn't fully resolve the issue because the replay buffer may still contain data collected by earlier versions of $\pi_\theta$, denoted $\pi_{\theta, t'}$, where $t'$ corresponds to early training steps. I also agree that in some games might have similar content in the buffer, therefore DFS performs similar to uniform sampling, which is what we have observed in the learning curves.
> > > >
> > > > Having said all this, we believe this is an interesting research direction that requires thoughtful solutions to address the outlined challenges, but it is beyond the scope of this paper. Thank you for the discussion and your valuable insights.

---

### Official Review · Reviewer_3CFw · 2024-11-03

**Soundness:** 3
**Presentation:** 3
**Contribution:** 3
**Rating:** 6
**Confidence:** 4

**Summary:**

This paper introduces DRAMA, a model-based reinforcement learning (MB-RL) agent that leverages the Mamba-2 architecture, a state space model (SSM), as its core dynamics architecture. Traditional MB-RL approaches often rely on recurrent neural networks (RNNs) or transformers for world modelling, which suffer from issues like vanishing gradients, difficulty in capturing long-term dependencies, and quadratic scaling of computational complexity with sequence length. DRAMA addresses these challenges by utilizing Mamba-2, which achieves linear computational and memory complexity while effectively capturing long-term dependencies.

Additionally, the authors propose a novel dynamic frequency-based sampling (DFS) method to mitigate the suboptimality arising from imperfect world models during early training stages. They evaluate DRAMA on the Atari100k benchmark, demonstrating that it achieves performance comparable to state-of-the-art algorithms using a significantly smaller world model (7 million parameters) that can be trained on standard hardware. The paper also includes ablation studies comparing Mamba-1 and Mamba-2, highlighting the superior performance of Mamba-2 despite its constrained expressive power.

**Strengths:**

- **Originality:** The paper introduces the novel application of Mamba-2 SSMs within MB-RL, specifically as the dynamics model in the world model. This is a new approach that addresses the limitations of existing architectures like typical RNNs and transformers and it makes a lot of sense in my opinion.
- **Quality:** The authors provide thorough experimental evaluations on the Atari100k benchmark, demonstrating that DRAMA achieves competitive performance with significantly fewer parameters (at least in the tasks and model sizes tested). The inclusion of ablation studies comparing Mamba-1 and Mamba-2, as well as the impact of DFS, strengthens the empirical results.
- **Clarity:** The paper is well-written and structured, providing clear explanations of the methodology, including detailed descriptions of Mamba-2 and how it is integrated into the world model. The figures and tables effectively support the textual content although I would make some further stylistic improvements in Figure 1 to maximise clarity.
- **Practical impact:** By achieving comparable performance to state-of-the-art methods with a smaller and more computationally efficient model, the DRAMA method contributes to making MB-RL more accessible and practical, particularly in resource-constrained environments.

**Weaknesses:**

- **Unsupported claims about capturing long-term dependencies:** While the authors claim repeatedly that Mamba-2 effectively handling long-term dependencies, the paper provides limited direct evidence or analysis to demonstrate this capability. Including experiments or analyses that specifically test and showcase the ability to capture long-term dependencies would strengthen the paper. For instance, a task designed to require long-term memory or metrics that quantify the model's ability to capture dependencies over long sequences could be included. In the current form of the paper, it is not clear to me what "long" really means.
- **Limited Comparison with Scaled Models:** The comparison with DreamerV3 is somewhat limited. While the authors emphasize parameter efficiency, it would be valuable to see how DRAMA performs when scaled up to match the model size of DreamerV3, even on a subset of games. This would help understand the limits and potential of their approach and whether the advantages of Mamba-2 persist at larger scales. If hardware limitations prevented this, a discussion of these constraints would be very helpful.
- **Marginal Performance Gains:** While DRAMA achieves comparable performance to existing methods, the improvements are not substantial across all games. Demonstrating scenarios where DRAMA significantly outperforms other approaches would strengthen the claims about its effectiveness.
- **DFS Method Clarity:** The explanation of the dynamic frequency-based sampling method could be more detailed. Providing more comprehensive comparisons with other sampling strategies would help in understanding its effectiveness. Additionally, including ablation studies that isolate the impact of DFS would clarify its contribution.
- **Hyperparameter Sensitivity:** The paper mentions that increasing the model size leads to better performance but does not deeply explore this aspect. Though I understand that extensive hyperparameter search might be too difficult given computational constraints, some analysis of how sensitive DRAMA is to hyperparameter choices, including model size, sequence length, and learning rates, would be very valuable for understanding its practical applicability, otherwise the paper looks relatively incomplete.

**Questions:**

- **Evidence of Long-Term Dependency Capture:** You mention the ability of DRAMA to capture long-term dependencies. Could you provide more direct evidence or experiments that demonstrate this capability? For example, have you considered tasks that specifically require long-term memory or conducted analyses that quantify the effective memory length of the model?
- **Scaling DRAMA for Direct Comparison:** Have you considered scaling up DRAMA to match the model size of DreamerV3 for a direct comparison? If not, could you explain the limitations (e.g., hardware constraints) that prevented this? Testing DRAMA with larger models on a few games could provide insights into its scalability and performance limits.
- **Effectiveness of DFS:** Could you elaborate on how the dynamic frequency-based sampling (DFS) method compares to other sampling strategies in terms of its impact on learning efficiency and final performance? Including quantitative comparisons or ablation studies would be helpful.
 - **Hyperparameter Sensitivity and Trade-offs:** Can you provide insights into the trade-offs between model size and performance in DRAMA? Specifically, how does increasing the size of the Mamba-2 model or the autoencoder affect results across different games?
- **Code and Reproducibility:** Is there a plan to release the code and pretrained models for DRAMA to facilitate reproducibility and further research in this area?

---

> ### Author Response · Authors · 2024-11-26
>
> > Unsupported claims about capturing long-term dependencies
>
> Our approach is motivated by evidence in the literature showing that State-Space Models (SSMs) have the ability to capture long-term dependencies, making them particularly effective for long-range modeling tasks, such as those in the Long Range Arena [Tay et al., 2021; Gupta et al., 2022; Smith et al., 2023]. Sequence lengths in this domain range from 1,024 to over 16,000. Mamba1 and Mamba2 inherit this capability as they are SSMs. Related work [Deng et al., 2023] has demonstrated that SSMs excel as world models, effectively capturing dynamics in specially tailored environments designed to measure long-term memory capabilities.
>
> To address the reviewer’s concern, we conducted an ablation study (in Sec 3.2.3) focusing on the critical components of the world model: the dynamics model. In a simple yet representative grid-world scenario, our results confirm that Mamba2 effectively captures long-term dependencies as expected. The long-term in the ablation experiment is refered as 1664 training sequence length. The result can be seen in **Table 2** above.
>
> **Reference**:
>
> Smith, Jimmy T. H., Andrew Warrington, and Scott W. Linderman. "Simplified State Space Layers for Sequence Modeling." In *The Eleventh International Conference on Learning Representations*, 2023.
>
> Tay, Yi, Mostafa Dehghani, Samira Abnar, Yikang Shen, Dara Bahri, Philip Pham, Jinfeng Rao, Liu Yang, Sebastian Ruder, and Donald Metzler. "Long Range Arena: A Benchmark for Efficient Transformers." In *International Conference on Learning Representations*, 2021.
>
> Gupta, Ankit, Albert Gu, and Jonathan Berant. "Diagonal State Spaces Are as Effective as Structured State Spaces." In *Advances in Neural Information Processing Systems*, 35:22982–22994, 2022.
>
> Deng, Fei, Junyeong Park, and Sungjin Ahn. "Facing Off World Model Backbones: RNNs, Transformers, and S4." In *Advances in Neural Information Processing Systems*, 36:72904–72930, 2023.
>
>
> > Scaling DRAMA for Direct Comparison
>
> Scaled model comparisons require substantial computational resources and time, which were not available to us. One key motivation for presenting Mamba-2 as a world model is its parameter efficiency. While scalability is undoubtedly important, smaller yet efficient models offer significant advantages. For instance, model-based reinforcement learning (MBRL) often faces the challenge of *model exploitation*, where the behavior model exploits imperfections in the world model to achieve higher rewards by repeatedly reaching states where the world model is underfitting. A potential solution to this issue is training multiple models to estimate uncertainty in predictions. However, this approach requires smaller and more efficient models, making Mamba-2 well-suited for such future directions.
>
> To specifically address concerns and provide a ‘like-for-like’ comparison, we trained a 12M version of DreamerV3 on the Atari100k benchmark and reported the results in the appendix. The results demonstrate that Mamba-2 achieved a significant advantage over this variant of DreamerV3 in the domain of small models on the Atari100k benchmark. The results are presented in **Table 1** above, with the detailed table and training curves provided in Appendix A.1 of the revised version.
>
> > DFS Method Clarity
>
> To address this concern, we conducted a study comparing uniform sampling and DFS in Drama across all games. The results demonstrate the effectiveness of DFS, which outperformed uniform sampling in 11 games, underperformed in 2 games, and tied in 13 games.
>
> > Hyperparameter Sensitivity
>
> We agree that this is an important concern. In our evaluation, we used the default hyperparameters for Mamba-2, while the other components of Drama were configured similarly to DreamerV3, with one exception: the actor was set to half the size of the critic. However, model-based RL inherently requires significant computational resources for hyperparameter tuning due to the complexity of its components, including the autoencoder, dynamics model, and behavior policy. Given these constraints, we prioritised using the available computational resources for the other requested ablation experiments instead. We plan to evaluate the sensitivity of the model to other hyperparameter values in future work.
>
> > Code and Reproducibility
>
> We will release the code repo once the anonymity is no longer applied.

---

> > ### Comment · Reviewer_3CFw · 2024-11-27
> >
> > Thank you for your responses and additional experimental results. The new ablation studies have addressed my main concerns:
> >
> > * The sequence length comparison between architectures helps substantiate your claims about long-term dependencies, though I suggest explicitly defining "long-term" in your context.
> > * The DramaXS vs DreamerV3XS comparison provides a convincing demonstration of parameter efficiency at smaller scales.
> > * The comprehensive DFS ablation across all games clarifies its contribution to the method's performance.
> >
> > Finally, while hyperparameter sensitivity remains underexplored due to computational constraints, I understand this limitation and appreciate your transparency about it.
> >
> > Given these improvements and clarifications, I am also slightly upgrading my rating. The paper makes a valuable contribution in demonstrating the effectiveness of Mamba-based world models with significantly fewer parameters.

---

> > > ### Author Response · Authors · 2024-11-28
> > >
> > > Thank you for your questions and advice. We appreciate your confirmation of our contribution.
> > >
> > > > ...though I suggest explicitly defining "long-term" in your context.
> > >
> > > We agree and have added a footnote in the introduction (at page 2 line 64) where we first introduce the phrase 'long-term'. The footnote states:
> > >
> > >
> > > 'According to (Tay et al., 2021), a long sequence is defined as having a length of 1,000 or more.'

---

### Author Response · Authors · 2024-11-26
**Revised Version Change Summary**

- **Comparison of DramaXS and DreamerV3XS**:
  To enable a like-for-like comparison between Drama and DreamerV3 with a similar number of parameters, we trained a version of Dreamer with only 12M parameters (referred to as DreamerV3XS) on the full Atari100K benchmark. The DramaXS model has 10M parameters in total (7M for the world model).

  | Metric                   | DramaXS | DreamerV3XS |
  |--------------------------|---------|-------------|
  | Normalised Mean Score    | 105     | 37          |
  | Normalised Median Score  | 37      | 7           |
  **Table 1**: Atari100k benchmark performance.

- **Additional Ablation Experiment on Long-Sequence Predictability Tasks**:
  We conducted an additional ablation experiment (Sec. 3.2.3) on long-sequence predictability tasks using widely used dynamic models in MBRL: Mamba1 (Drama), Mamba2 (Drama), Transformer (IRIS, TWM, STORM), and GRU (Dreamer). Both Mamba1 and Mamba2 demonstrated equivalent strong performance while maintaining shorter training times.

    | **Method**        | **$l$**   | **Training Time (ms)** | **Memory Usage (%)** | **Error (%)**     |
    |--------------------|---------|------------------------|-----------------------|--------------------|
    | **Mamba-2**       | 208     | 25                     | 13                    | 15.6 ± 2.6         |
    |                   | 1664    | 214                    | 55                    | 14.2 ± 0.3         |
    | **Mamba-1**       | 208     | 34                     | 14                    | 13.9 ± 0.4         |
    |                   | 1664    | 299                    | 52                    | 14.0 ± 0.4         |
    | **GRU**           | 208     | 75                     | 66                    | 21.3 ± 0.3         |
    |                   | 1664    | 628                    | 68                    | 34.7 ± 25.4        |
    | **Transformer**   | 208     | 45                     | 17                    | 75.0 ± 1.1         |
    |                   | 1664    | -                      | OOM                   | -                  |
    **Table 2**: Performance comparison of different methods on the grid world environment.

- **Extended DFS Uniform Ablation Experiments**:
  We extended the DFS uniform ablation experiments to the full Atari100k benchmark as requested. The results show that DFS demonstrated its effectiveness by outperforming in 11 games, underperforming in 2 games, and achieving similar performance (within a 5% margin) in 13 games. This indicates that DFS is effective when combined with Drama. The detailed learning curves and table can be found in the Appendix A.2 of the revised version.

  | Metric                   | DFS | Uniform |
  |--------------------------|---------|-------------|
  | Normalised Mean Score    | 105     | 80          |
  | Normalised Median Score  | 37      | 28           |

  **Table 3**: Atari100K Benchmark Performance: DFS vs. Uniform Sampling with Drama XS Model.

---

### Meta-Review · Area_Chair_VF4P · 2024-12-23

**Metareview:**

The paper presents a significant contribution by effectively incorporating Mamba architecture into model-based RL, achieving competitive performance with only 7M parameters. While reviewers raised concerns about long-term dependency claims and computational efficiency, the authors provided comprehensive responses with new experimental results demonstrating Mamba's effectiveness at long sequences, faster imagination time than transformers, and improved DFS performance in 11 games. The authors also conducted thorough ablations showing superior performance over small-scale DreamerV3. With all reviewers responding positively to these detailed responses, and given the practical impact of achieving state-of-the-art performance with significantly reduced parameters, this work warrants acceptance.

**Additional Comments On Reviewer Discussion:**

The reviewers raised concerns about the model's performance, computational efficiency, and experimental validation. The authors responded by conducting additional ablation studies, providing wall-clock training time comparisons, clarifying notational issues, and expanding theoretical motivations for the Mamba architecture. Reviewers found these responses satisfactory, with most increasing their scores and appreciating the novel approach to model-based reinforcement learning. The discussion emphasized the paper's potential to make reinforcement learning more accessible through a parameter-efficient method, ultimately leading to a consensus on the work's valuable contribution.

---

### Decision · Program_Chairs · 2025-01-22

Accept (Poster)